# JP-14: A Trace Amine-Associated Receptor 1 Agonist with Anti-Metabolic Disorder Potential

**DOI:** 10.3390/ijms262010033

**Published:** 2025-10-15

**Authors:** Monika Marcinkowska, Joanna Sniecikowska, Monika Głuch-Lutwin, Barbara Mordyl, Marek Bednarski, Adam Bucki, Michał Sapa, Monika Kubacka, Agata Siwek, Agnieszka Zagórska, Jacek Sapa, Marcin Kołaczkowski, Magdalena Kotańska

**Affiliations:** 1Department of Medicinal Chemistry, Faculty of Pharmacy, Jagiellonian University Medical College, 9 Medyczna Street, 30-688 Krakow, Poland; monika.marcinkowska@uj.edu.pl (M.M.); joanna.sniecikowska@uj.edu.pl (J.S.); adam.bucki@uj.edu.pl (A.B.); michal.piotr.sapa@doctoral.uj.edu.pl (M.S.); agnieszka.zagorska@uj.edu.pl (A.Z.); marcin.kolaczkowski@uj.edu.pl (M.K.); 2Department of Pharmacobiology, Jagiellonian University Medical College, 9 Medyczna Street, 30-688 Krakow, Poland; monika.gluch-lutwin@uj.edu.pl (M.G.-L.); barbara.mordyl@uj.edu.pl (B.M.); agat.siwek@uj.edu.pl (A.S.); 3Laboratory of Pharmacological Screening, Department of Pharmacodynamics, Jagiellonian University Medical College, 9 Medyczna Street, 30-688 Krakow, Poland; marek.bednarski@uj.edu.pl (M.B.); monika.kubacka@uj.edu.pl (M.K.); jacek.sapa@uj.edu.pl (J.S.)

**Keywords:** trace amine-associated receptor 1 agonist, TAAR1, lipid metabolism, gastric emptying, aminoguanidine-based ligand

## Abstract

TAAR1 agonists have emerged as promising therapeutic agents capable of modulating glucose homeostasis, enhancing insulin secretion and suppressing appetite, making them attractive candidates for the treatment of obesity and related metabolic disorders. Despite their potential, the number of TAAR1-targeting compounds with well-defined pharmacological profiles remains limited. In this study, we identified and characterized JP-14, a novel aminoguanidine-based TAAR1 agonist, in a comprehensive panel of pharmacological assays. JP-14 promoted glucose uptake in HepG2 cells and reduced lipid deposition during 3T3-L1 adipocyte differentiation, with both actions dependent on TAAR1 signaling. In differentiated 3T3-L1 adipocytes, JP-14 reduced intracellular levels of both neutral lipids and phospholipids, indicating dual anti-steatotic and anti-phospholipidotic activity. In zebrafish larvae, toxicity profiling confirmed 10 µg/mL as a safe concentration for further in vivo studies. These assays showed that JP-14 promoted lipid mobilization and partially prevented fructose-induced lipid accumulation, demonstrating systemic metabolic benefits in vivo. Moreover, JP-14 markedly delayed gastric emptying in mice, an effect similar to loperamide and reversed by TAAR1 antagonism, supporting its role in regulating satiety and energy balance. Collectively, our findings establish JP-14 as a safe and metabolically active TAAR1 agonist with multifaceted effects on glucose and lipid metabolism. JP-14 represents a valuable pharmacological tool for probing TAAR1-mediated mechanisms in metabolic regulation.

## 1. Introduction

Trace amines (TAs) are naturally occurring molecules found at very low levels across different tissues that were long regarded as inert byproducts of monoamine neurotransmitters such as dopamine and serotonin. Later, they were recognized to exert modulatory effects through trace amine-associated receptors (TAARs), a class of G protein-coupled receptors (GPCRs), with the TAAR1 subtype being the most extensively characterized [1]. TAAR1 is widely expressed in both the central nervous system and peripheral tissues, where it modulates key physiological processes, including neurotransmission, metabolism, and cardiovascular function [2]. Beyond its physiological role, TAAR1 has been implicated in pathological processes underlying metabolic disorders, including obesity and metabolic syndrome. TAAR1 is expressed in pancreatic islet β-cells, where it plays a key role in regulating insulin secretion. In individuals with metabolic syndrome, TAAR1 expression and its functional interactions appear to be dysregulated, suggesting its involvement in metabolic dysfunction [3]. Genetic evidence shows that naturally occurring TAAR1 variants have been identified in individuals with obesity and impaired glucose homeostasis. Certain mutations in the TAAR1 gene, such as Arg23Cys [4], lead to a complete loss of insulin production, highlighting a direct role for TAAR1 dysfunction in metabolic disturbances. Interestingly, these dysfunctions may be mitigated by TAAR1 ligands, as activation of TAAR1 by agonists has been shown to improve glycemic control and promote body weight in rodent models of obesity and diabetes. In particular, TAAR1 activation has been reported to reduce fasting glucose levels, delay gastric emptying, enhance satiety, and suppress food intake, effects that highlight its therapeutic relevance for obesity management [2]. Together, these findings have driven research toward the therapeutic potential of TAAR1 as a molecular target for the treatment of metabolic disturbances [5].

Despite the high promiscuity of TAAR1 ligands and their potential therapeutic utility, the number of reported TAAR1 ligands remains limited, which highlights the need for further investigation in this area. The identification and characterization of novel TAAR1 binders would not only enhance our understanding of the physiological role of TAAR1 but also provide valuable insights for the development of future therapeutics targeting this receptor. Herein, we present a novel TAAR1 ligand (JP-14) based on an aminoguanidine chemical scaffold and provide an in-depth evaluation of its pharmacological properties, with a particular focus on its potential application in the treatment of obesity and metabolic disturbances.

## 2. Results and Discussion

### 2.1. Design, Synthesis, and Identification of Compound JP-14 as Novel TAAR1 Ligand

Among the few chemotypes reported as TAAR1 ligands, we focused on the aminoguanidine scaffold, given that analogs such as guanabenz [6] and its 4-hydroxy metabolite—4-OH-guanabenz [7]—have previously been shown to exhibit high affinity for TAAR1 (EC_50_ = 8.5 nM and EC_50_ = 316.3 μM). The aminoguanidine moiety, presented in these compounds, mimics the interaction of amine functions in trace amines with aspartate residues (Asp68, Asp102, and Asp284) in TAAR1, which are critical for ligand recognition [7,8]. However, despite their promising affinities toward TAAR1, both compounds exhibit substantial affinity for the α_2_-adrenoceptor (*K*_i_ = 2.6 nM and 19.4 nM, Table 1), which may lead to undesirable side effects. Given the therapeutic potential for treating metabolic disorders, it would be desirable to identify selective TAAR1 ligands that do not interact or exhibit limited interaction with α_2_-adrenoceptors, to minimize potential side effects (headache and dizziness, drowsiness and fatigue, heart rhythm and blood pressure disturbances, etc.). We envisioned that guanabenz could serve as a starting point for further modifications aimed at reducing α_2_-adrenoceptor affinity while enhancing binding to TAAR1.

In our previous work, we observed that modifying the substitution pattern on the aromatic ring could substantially influence and reduce binding to α-adrenergic receptors, for example, by introducing additional substituents at the 3,4-positions of the phenyl ring. In addition, we observed that literature examples of structurally related biguanide derivatives, such as BIG12–BIG16, featuring substituents at the 3- and 4-positions of the aromatic ring [9,10], have been reported to exhibit particularly strong activity toward TAAR1 [11]. Based on these findings, we incorporated similar modifications into our series by introducing various substituents at the 3- and 4-positions of the aromatic ring while retaining the aminoguanidine moiety as the core structural feature essential for interactions with the Asp residue of TAAR1 (Figure 1).

A focused library of aminoguanidine derivatives was synthesized and evaluated for agonist and antagonist activity at human TAAR1 using a cAMP accumulation assay (Appendix A). Additionally, off-target activity was assessed at the 5-HT_2C_, α_1_-, and α_2_-adrenoceptors. Among the synthesized library of aminoguanidine derivatives, compound JP-14 (Figure 2) emerged as the most promising candidate, exhibiting notable agonist activity at TAAR1, with an EC_50_ value of 11.29 ± 1.4 μM (Table 1). Although its potency was slightly lower than that of the reference agonist tyramine (EC_50_ = 5.87 ± 0.47 μM), JP-14 demonstrated consistent and measurable receptor activation.

Interestingly, the maximum cell stimulation (secondary messenger level, cAMP) observed in the presence of the compound JP-14 was about twice as high as the maximum cell stimulation in response to the highest concentration of tyramine (Figure 3). This may suggest that the JP-14 compound increases nonspecifically the level of cAMP in the cell, independently of TAAR1 stimulation. This effect may result either from increased cAMP production, e.g., as a result of stimulation of adenylate cyclase, as in the case of forskolin, or from inhibition of cAMP-degrading enzymes such as phosphodiesterases. Further studies will be necessary to understand this mechanism.

In addition to the compound JP-14, guanabenz also demonstrated agonistic properties toward TAAR1. The EC_50_ value was 0.3 µM, and it induced maximal receptor activation at 50% of the signal produced by the full agonist, tyramine. Functional studies indicated that guanabenz is a partial agonist of TAAR1.

Considering that in heterologous cell lines, β-phenylethylamine and para-tyramine stimulate multiple species TAAR1 (mouse, rat, and human) with reported EC_50_ values ranging from 0.1 to 1 μM, and that tryptamine and para-octopamine show broader activity ranges (EC_50_ = 0.4–21 μM and 2–20 μM, respectively) [12], the potency and efficacy of our most active compound, JP-14, are particularly encouraging. The calculated EC_50_ value and robust cellular response in TAAR1-expressing cells position JP-14 as a promising lead for further pharmacological development.

To further understand the molecular interactions of JP-14 with TAAR1, molecular modeling studies were performed to elucidate the specific binding mode and key contact residues involved in receptor engagement. Due to restricted rotation around the C=N bond, the aminoguanidine moiety present in the structure of JP-14 may adopt E and Z configurations, potentially giving rise to two distinct stereoisomers with different interaction profiles (Figure 4). Therefore, we performed quantum mechanical calculations using the Jaguar module to determine the lowest-energy conformations of the E and Z isomers of JP-14. The calculated conformational energies for the E- and Z-JP-14 isomers were −67.72 kcal/mol and −64.59 kcal/mol, respectively. The lower energy of the E isomer suggests that it adopts a more stable conformation, which may favor its binding to the TAAR1 receptor. Molecular docking studies confirmed the characteristic binding pattern of TAAR1 ligands [13], as exhibited by the E isomer, which also received a more favorable Glide GScore compared to the Z isomer, suggesting a stronger predicted binding affinity for the TAAR1 receptor. Moreover, E-JP-14 demonstrated a more consistent docking pose than the Z isomer. Its guanidine group formed both an ionic bond and a hydrogen bond with Asp3.32, and this group was further stabilized by a π-cation interaction with Phe6.51. Moreover, the lone electron pair of the nitrogen atom in the dichloropyridine ring is permanently localized and does not undergo conformational changes, as observed in the case of 4-hydroxyguanabenz. This may contribute to greater stability of the bond and, consequently, to an increase in functional activity of JP-14.

There are examples of TAAR1 ligands in the literature that concurrently bind to serotonin 5-HT_2C_ receptors [14], which, theoretically, if they were centrally acting agonists, could be beneficial in terms of reducing obesity [15]. However, each additional mechanism of action is also associated with a greater risk of adverse effects. Our further analysis of the off-target profile of our lead compound JP-14 revealed that it did not exhibit any measurable agonistic or antagonistic activity at the 5-HT_2C_ receptor, as determined by intrinsic activity assays (Table 1). This absence of functional interaction suggests a favorable selectivity profile for JP-14, minimizing the risk of serotonergic side effects and supporting its potential utility in TAAR1-targeted therapeutic applications (Table 1).

Our study shows that JP-14 acts as a weak partial agonist at the α_2A_-adrenoceptor, inducing approximately 15% receptor activation relative to the full agonist control (Table 1). In physiological body conditions, catecholamines target the postsynaptic α_2A_-adrenoceptor found on pancreatic islet β-cells, stopping the release of insulin, which leads to a rise in blood glucose levels [16]. Therefore, lowering the activity of the α_2A_-adrenoceptor could help promote insulin release from pancreatic islets and, in turn, decrease high glucose levels [17]. In our previous study, we demonstrated that guanabenz—a full α_2A_-adrenoceptor agonist (EC_50_ = 16.32 nM) [7]—in an unexpected observation, normalized the glucose levels raised by obesity in rats [18]. Of note, these findings were confirmed also by other studies [19,20]. In subsequent studies, we demonstrated that the guanabenz metabolite 4-OH-guanabenz, a partial agonist activating the α_2A_-adrenoceptor by 30% (EC_50_ = 316.3 nM), also beneficially affected the lipid and carbohydrate profile in obese rodents [7]. Given that JP-14 is a weak partial agonist of the α_2A_-adrenoceptor, eliciting only 15% receptor activation, it is unlikely to negatively impact carbohydrate metabolism in obese individuals—though this warrants future investigation. Moreover, its partial agonism at α_2A_-adrenoceptors may theoretically exert a modulatory effect on insulin resistance (due to hyperinsulinemia), giving a rest to pancreatic islets by reducing excessive insulin secretion. This hypothesis, however, requires further experimental validation. However, it is highly likely that other mechanisms of action of these compounds, overriding α_2A_-adrenoceptor activation, could explain the lower levels of plasma glucose, and this is very clear and supported by the available evidence.

In parallel, in our study, JP-14 did not exhibit notable affinity for α_1_-adrenergic receptors, suggesting a low risk of cardiovascular side effects associated with α_1_-adrenergic engagement. Further analysis revealed that JP-14 exhibited weak antagonistic activity at the α_2B_-adrenoceptor, with an IC_50_ value of 622 nM. Nevertheless, given its favorable profile—robust TAAR1 agonism, no interaction with the 5-HT_2C_ or α_1_-adrenoceptors, and only modest effects at α_2_ subtypes of adrenoceptors—JP-14 was deemed suitable for further pharmacological evaluation.

### 2.2. Biological Activity in In Vitro and In Vivo Assays

#### 2.2.1. Influence of JP-14 on Glucose Metabolism in HepG2 Cells

The effect of JP-14 on glucose consumption was evaluated using HepG2 cells, a widely accepted in vitro model for hepatic glucose metabolism, to investigate the metabolic activity. Metformin (100 µM), a clinically established agent facilitating glucose uptake into cells (an anti-hyperglycemic agent), was included as a positive control. It significantly increased glucose uptake by 36% relative to vehicle-treated cells, thereby validating the sensitivity and reliability of the assay. Among the tested compounds, tyramine (a known TAAR1 agonist, 10 µM), and JP-14 (10 µM) elicited statistically significant increases in glucose consumption, enhancing uptake by 32%, and 29%, respectively. These effects were comparable in magnitude to those observed with metformin. In contrast, no significant changes in glucose uptake were observed for RTI-7470-44 (10 µM, antagonist of TAAR1), meta-chlorophenylpiperazine (mCPP—ligand of numerous receptors, including **α**_2_-adrenergic, serotonergic receptors (mainly 5-HT_2C_) and TAAR1; 10 µM). The results are shown in Figure 5.

To confirm that the stimulation of glucose consumption by JP-14 is mediated through TAAR1, follow-up experiments were conducted in the presence of the selective TAAR1 antagonist RTI-7470-44 (40 nM). Glucose uptake assays were repeated for both JP-14 and tyramine under conditions of pharmacological TAAR1 blockade. The presence of RTI-7470-44 substantially attenuated the glucose-stimulating effects of JP-14 and tyramine (Figure 5), supporting the hypothesis that TAAR1 contributes mechanistically to the metabolic activity of JP-14. These findings further validate the therapeutic potential of TAAR1 in modulating hepatic glucose metabolism.

#### 2.2.2. Influence of JP-14 on Adipocyte Lipid Accumulation in 3T3-L1 Cells

To further delineate the metabolic effects of JP-14, its impact on lipid accumulation during adipocyte differentiation was evaluated using the murine 3T3-L1 cell line, a commonly utilized in vitro model for examining adipogenesis and lipid metabolism. Rosiglitazone (100 µM) was included as a positive control. It significantly decreases lipid accumulation by 90% relative to vehicle-treated cells, thereby validating the sensitivity and reliability of the assay. It was observed that tyramine (10 µM) and JP-14 (10 µM) significantly reduced lipid accumulation, lowering intracellular lipid content by 33% and 35%, respectively, compared to vehicle-treated controls (Figure 6). Surprisingly, mCPP (10 µM) significantly increased lipid accumulation. This may undoubtedly be related to a mechanism of action other than TAAR1, most plausibly its action on serotonin receptors, for which mCPP is a known ligand.

To determine whether the anti-lipogenic effects of tyramine and JP-14 were mediated via TAAR1, follow-up experiments were conducted in the presence of the selective TAAR1 antagonist RTI-7470-44 (40 nM). Notably, co-treatment with RTI-7470-44 completely abolished the lipid-lowering effects of both tyramine and JP-14 (Figure 6). These findings provide strong evidence that tyramine and JP-14 inhibit lipid accumulation in differentiating adipocytes through a TAAR1-dependent mechanism. The complete loss of activity in the presence of the antagonist indicates that receptor engagement is essential for their biological effects in this context.

Ligands targeting TAAR1 and α_2A_-adrenoceptors have previously been reported to exert effects similar to those of JP-14, notably enhancing hepatic glucose consumption and reducing lipid accumulation in adipose tissue. These include, for example, endogenously occurring 3-Iodothyronamine and its synthetic derivatives [21], proposed some years ago as interesting in terms of the potential to combat obesity. These compounds cause alterations in metabolism that mainly involve a change in the way the body uses nutrients, moving from burning carbohydrates to lipids and hindering lipogenesis [22], which leads to a significant reduction in body weight in animals [23,24].

Furthermore, the dual activity of JP-14 enhancing glucose consumption in hepatocytes and simultaneously reducing lipid accumulation in adipocytes positions it as a particularly promising lead compound for metabolic modulation. The specificity of these effects and their reversibility upon TAAR1 antagonism underscore the receptor’s main role in mediating both hepatic and adipose responses.

#### 2.2.3. Influence of Tested Compound on Steatosis and Phospholipidosis in 3T3-L1 Adipocytes

High-content imaging analysis was conducted to evaluate the effects of selected compounds on intracellular lipid accumulation in differentiated 3T3-L1 adipocytes. This analysis employed an assay that enables simultaneous visualization and quantification of neutral lipid (steatosis) and phospholipid (phospholipidosis) accumulation at the single-cell level (Figure 7). This method is critical for accurately assessing compound-specific modulation of lipid metabolism and for identifying potential anti-steatotic or anti-phospholipidotic agents in adipocyte models. High-content imaging results showed that the selected compounds significantly reduced intracellular neutral lipid accumulation in 3T3-L1 adipocytes.

As shown in Figure 8, both tyramine (10 µM) and JP-14 (10 µM) caused a statistically significant decrease in steatosis compared to treatment with vehicle alone. Importantly, co-treatment with RTI-7470-44 (40 nM), a selective TAAR1 antagonist, abolished the suppressive effects of all tested compounds on steatosis. No significant variations were seen between the groups treated with RTI-7470-44 and the control group, indicating that the reduction in steatosis is likely mediated via TAAR1-dependent signaling.

Similarly, in the phospholipidosis assay, JP-14 (10 µM) and tyramine (10 µM) significantly reduced phospholipid accumulation in 3T3-L1 cells, with the strongest effects observed for JP-14. As with steatosis, co-incubation with RTI-7470-44 (40 nM) reversed these effects (Figure 9). In the presence of TAAR1 blockade, none of the tested compounds significantly affected phospholipidosis levels compared to control, further supporting the involvement of TAAR1 signaling in the regulation of phospholipid accumulation in this cellular model.

It is important to note that while both steatosis and phospholipidosis involve intracellular lipid accumulation, they represent distinct lipid storage pathologies with different cellular implications. Steatosis is characterized by the accumulation of neutral lipids, mainly triglycerides, and is often associated with metabolic dysfunction, such as insulin resistance, lipotoxicity, and inflammation, particularly in adipose and hepatic tissues [25]. In contrast, phospholipidosis involves the intracellular accumulation of phospholipids, often within lysosomal compartments, and is frequently linked to drug-induced toxicity and impaired lipid turnover or degradation. Persistent phospholipidosis can interfere with normal lysosomal function, membrane dynamics, and cell signaling [26].

Specifically, chronic drug-induced phospholipidosis is known to disturb lysosomal function by impeding normal lipid degradation and altering membrane behavior. For instance, cationic amphiphilic drugs such as amiodarone cause the accumulation of numerous intra-lysosomal lamellar bodies (myeloid inclusions) in cells, indicative of impaired phospholipid catabolism similar to lysosomal storage disorders [26]. These persistent phospholipid accumulations alter the lysosomal membrane environment and neutralize membrane charge, which interferes with the binding and activity of lipid-degrading enzymes and attenuates normal lipid turnover. Consequently, lysosomal enzymes can become mislocalized or degraded, and undegraded lipids build up, interfering with membrane dynamics (e.g., vesicle formation and fusion events) and downstream cellular processes. Indeed, cells undergoing amiodarone-induced phospholipidosis show clear signs of lysosomal dysfunction—including the leakage of hydrolases, incomplete maturation of cathepsins, and perturbation of signaling pathways dependent on lysosomal homeostasis. Notably, in phospholipidotic cells, the nutrient-sensing mTORC1 pathway is suppressed, while compensatory activation of autophagy regulators such as TFEB is observed [27]. These examples illustrate how persistent phospholipid overload can interfere with normal lysosomal function, membrane dynamics, and cell signaling.

Therefore, the ability of the tested compounds to reduce both steatosis and phospholipidosis, in a TAAR1-dependent manner, suggests a potentially beneficial dual mechanism, which may have implications for developing therapies targeting lipid storage disorders and metabolic diseases such as obesity, type 2 diabetes, and lipotoxic organ damage.

Considering the comprehensive data derived from receptor binding assays and functional cellular studies, including JP-14’s robust and TAAR1-dependent inhibition of both lipid accumulation (steatosis) and phospholipidosis in 3T3-L1 adipocytes, we identified JP-14 as a compound with a favorable metabolic profile. Its dual capacity to modulate key aspects of adipocyte lipid homeostasis and hepatic glucose uptake underscores its therapeutic potential. Consequently, JP-14 was planned for further in-depth pharmacological evaluation.

#### 2.2.4. Influence of JP-14 on Lipid Utilization in Zebrafish Model

We investigated the effect of JP-14 on lipid utilization in the early developmental stage of zebrafish. In the fish yolk sac, the depletion of lipid occurs after embryos hatch [28]. During the early stages of zebrafish development, the yolk sac is the only source of energy for the embryo and larva, and it is a finite source of energy. Examination of yolk sac changes makes it possible to detect changes in the body’s lipid metabolism [29]. These drugs/compounds that accelerate the utilization of lipids from the yolk sac are considered to have a beneficial effect on lipid metabolism in the body [29], and the model of the effect on lipid utilization is used as a screening model to search for compounds/extracts that act favorably in this regard and potential drug targets for treating obesity [30].

Incubation of larvae with JP-14 at a concentration of 10 µg/mL for 48 h led to a decrease in Nile red detectable fat (Figure 10). A slightly stronger effect was observed after incubation of larvae with the reference compound—resveratrol at a concentration of 20 µg/mL. The concentration of JP-14 for this assay, i.e., 10 µg/mL was determined after the toxicity tests described below (see Section 2.4.3). These studies therefore clearly indicate a beneficial effect of JP-14 on lipid metabolism in a living organism.

#### 2.2.5. Influence of JP-14 on Fructose-Induction Lipid Disturbance in Zebrafish Model

Fructose is thought to be a major contributor to chronic metabolic illnesses, including obesity, hyperglycemia, insulin resistance, type 2 diabetes, hypertriglyceridemia, and non-alcoholic fatty liver disease [31]. It raises triglyceride levels and promotes fat accumulation in the liver and visceral tissues. Fructose is virtually totally eliminated by the liver. Hepatic metabolism of fructose stimulates lipogenesis [32]. These processes occur independently of insulin exertion and the phosphofructokinase regulatory phase. The process of adding a phosphate group to fructose in the liver uses up ATP, so the rising levels of ADP serve as a material for making uric acid. These actions lead to damage from oxidation in the liver e.g., lipid peroxidation. Fructose increases de novo lipogenesis, which leads to the development of nonalcoholic steatohepatitis or nonalcoholic fatty liver disease [33].

Zebrafish models of lipid disorders and hepatic steatosis have been previously established as effective screening models for evaluating the potential of drug candidates to modulate lipid metabolism and related pathologies in a living organism [34]. It has been discovered that the absorption of yolk can be hindered if the liver is damaged, so the size of the yolk has been proposed as a measure for how well zebrafish livers are working [35], and liver injury is associated with lipid accumulation in zebrafish [36].

Incubation of zebrafish larvae with 4% fructose solution induced significant yolk enlargement in these individuals. The compound JP-14 (10 µg/mL), with which the zebrafish larvae were treated in combination with fructose, partially attenuated fructose-induced yolk enlargement (Figure 11). Comparable effects were observed in the group of larvae treated with resveratrol (20 µg/mL). These observations were confirmed spectrophotometrically by quantifying the color intensity of the larval homogenates. Our preliminary research shows the beneficial effect of the JP-14 compound on interfering with lipid metabolism in the body.

#### 2.2.6. Influence of JP-14 on Gastric Emptying (Mouse Model)

Prior to in vivo evaluation, in silico prediction of drug-like properties was performed, indicating that the lead compound JP-14 exhibits high predicted oral bioavailability and shares key pharmacokinetic characteristics with guanabenz, an FDA-approved drug (see Section 2.3, below). These analyses, together with the pharmacological data described above, provided the rationale for conducting advanced in vivo studies with JP-14 using a mouse model to evaluate its effect on gastric emptying.

The gastric emptying model was selected as the primary in vivo proof-of-concept study based on the well-established role of TAAR1 in regulating gastrointestinal function. Gastric accommodation and emptying, which refers to how stomach content moves from the stomach to the intestines, play a key role in managing stomach stretching and how nutrients enter the intestines. This process helps regulate feelings of fullness and is also significant for maintaining a healthy body weight over time [37]. TAAR1 is expressed in the stomach, where its activation modulates key metabolic processes, including gastric motility and glucose homeostasis [2,38]. By delaying gastric emptying, agonists of TAAR1 can reduce the rate of glucose entry into circulation, improves glucose tolerance, and contributes to weight regulation and overall metabolic benefits. Gastric motility is an important modulator of hunger and satiety.

JP-14 administered intragastrically at a dose of 10 mg/kg body weight (b.w.) significantly inhibited gastric emptying in mice (Figure 12). The percentage inhibition of emptying was about 66%. This effect was comparable to the effect observed after intragastric administration of loperamide at a dose of 10 mg/kg b.w., which inhibited gastric emptying by about 55%. No statistically significant effect was determined in the other groups, but the combined administration of JP-14 with the TAAR1 inhibitor RTI-7470-44 (5 mg/kg b.w.) reduced the effect of JP-14 by about 40%—the percentage inhibition of gastric emptying in the group treated with both JP-14 and RTI-7470-44 was calculated to be about 25%. RTI-7470-44 alone did not affect emptying; the amount of dye measured in the stomach of mice treated with RTI-7470-44 alone was comparable to the amount of dye measured in the control group 30 min after dye administration. Metformin (100 mg/kg b.w.) had an inhibitory effect of about 30%.

Our study is consistent with recently published studies showing that other known TAAR1 agonists, including ulotaront or RO5166017, delay gastric emptying in mice after intragastric administration, thus beneficially affecting metabolism and thus potentially reducing body weight [5,39]. Delaying gastric emptying is a mechanism implicated in the weight-lowering effects of glucagon-like peptide-1 receptor agonists [40,41]. The management of food intake is partly regulated through gastric motor activities; delaying gastric emptying may assist weight loss by promoting satiety and decreasing calorie consumption [42]. Our previous studies demonstrated that guanabenz and its 4-hydroxy metabolite, both TAAR1 agonists, inhibit gastric emptying, significantly reduce body weight in obese animals, and exert favorable effects on glucose tolerance and lipid profile normalization [7,18]. The screening studies described here indicate JP-14 as a favorable candidate for further studies in the fight against obesity and metabolic disorders.

### 2.3. Chemical, Pharmacokinetic, and Drug-Likeness Properties

The evaluation of chemical, pharmacokinetic, and drug-likeness properties is an important component to consider when designing new small compounds endowed with biological activities, already at an early stage of drug discovery pipeline. We used the SwissADME and the web tools [43,44] to estimate critical parameters such as water solubility, drug-likeness, and pharmacokinetic properties for JP-14 and guanabenz (see Table 2 and Figure 13). The performed calculations indicate that both compounds have druglike properties according to Lipinski’s rules, have good synthetic accessibility scores, and are soluble in water. The sum of surface polar fragments, i.e., topological polar surface area (TPSA), is a good parameter for predicting drug transport properties. JP-14 and guanabenz have TPSA values equal to 87.15 Å^2^ and 76.76 Å^2^, respectively, which is a good range for oral bioavailability. Most therapeutic molecules have TPSA values of <140 Å^2^ in a non-polar environment and good passive membrane permeability [45].

The bioavailability radar is a tool for rapid evaluation of drug similarity—the results for a molecule should fall in a pink area so that it should be classified as drug-like. Figure 13a,b) show six physicochemical parameters: size, polarity, solubility, saturation, flexibility, and lipophilicity for JP-14 and guanabenz. The appropriate values are: SIZE—molecular weight between 150 and 500 g per mole; polarity (POLAR)—topological polar surface area (TPSA) ranging from 20 to 130 square angstroms (Å2); insolubility (INSOLU)—solubility measured by Log S (ESOL) from −6 to 0; insaturation (INSATU)—the fraction of carbon atoms with sp3 hybridization (fraction Csp3) between 0.25 and 1; flexibility (FLEX)—the count of rotatable bonds from 0 up to 9; lipophilicity (LIPO)—Log P (XLOGP3) values ranging from −0.7 to 5.0 [43]. Bioavailability radars for JP-14 and guanabenz exhibited suitable parameters in terms of size, polarity, insolubility, flexibility, and lipophilicity, but were only not good at insaturation.

The predicted human intestinal absorption and BBB penetration for JP-14 and guanabenz are illustrated using the forecast of boiled egg plots (Figure 13c). Both compounds are in the white box, indicating predictably good oral bioavailability and no ability to cross the BBB.

However, our previous in vivo studies using a rat model have shown that guanabenz is able to cross the blood–brain barrier after intraperitoneal administration [7]. In those studies, guanabenz was noted for its strong ability to pass through the BBB, having a brain-to-plasma ratio of 3.9. The entry of guanabenz into the brain was slow and happened within 60 min (t_max_), reaching its highest level of 183.2 ng/g. Another compound similar in structure to JP-14 and guanabenz, studied by our team earlier, i.e., 4-OH-guanabenz, brain permeability was markedly smaller than permeability of guanabez, while the entry of 4-OH-guanabenz into the brain occurred quickly (t_max_ = 15 min), yet it reached a lower peak level of 64.5 ng/g. Such knowledge emphasizes the need for additional studies on the possibility of crossing the BBB with the JP-14 compound.

Additionally, JP-14 and guanabenz are not defined as substrates for P-glycoprotein, which is advantageous from the point of view of their potential use as drugs [54]. P-glycoprotein is present in small amounts in many tissues, but it occurs in much greater quantities on the apical surface of the epithelial cells lining the small and large intestines, in the bile ducts of the liver, and in the proximal tubules of the kidneys, which serve excretory functions. Furthermore, it is a major component of the blood–brain barrier, transporting substrates into the blood, and is thus a significant factor limiting their entry into the brain [55]. P-glycoprotein is an efflux transporter that can pump drugs out of cells, which may cause a failed treatment [54]. P-glycoprotein influences the distribution of many administered drugs and plays a crucial role in the absorption, distribution, metabolism, and excretion (ADME) process. In the intestine, P-glycoprotein is responsible for the excretion of drugs into the intestinal lumen, thereby reducing their absorption and oral bioavailability [55]. Next, if a drug is identified as a substrate of this transporter, this means that its concentration levels in cells may be clinically affected by its inhibitors [54]. P-glycoprotein can interact with a wide range of structurally diverse compounds. Most of these are weakly amphipathic and relatively hydrophobic. Inhibition of P-glycoprotein results in increased absorption of substrates from the gastrointestinal tract and significantly slower elimination from the circulation, leading to dramatic increases in toxicity for some drugs. Furthermore, coadministration of two drugs that are P-glycoprotein substrates can lead to significant pharmacokinetic effects because they compete for the transporter. For these reasons, the Food and Drug Administration (FDA) currently recommends testing drugs for interactions with P-glycoprotein as part of the approval process [55].

### 2.4. Toxicity Studies

#### 2.4.1. Predicting the Most Frequently Assessed Toxicity Potentials with In Silico Models

The toxicity of JP-14 and guanabenz was predicted using the pkCSM (http://biosig.unimelb.edu.au/pkcsm/prediction, accessed on 15 July 2024) web tool [44]. Toxicity prediction for the compound JP-14 showed that it has similar toxicity to guanabenz (Table 3). The most important parameters are the lack of hepatotoxicity and the lack of inhibition of the human Ether-à-go-go-Related Gene (hERG, that codes the alpha subunit of a potassium ion channel).

#### 2.4.2. Preliminary Cytotoxicity Assessment of Test Compounds

Prior to investigating the biological activity and molecular characteristics of the test compounds in vitro, a comprehensive cytotoxicity screening was performed using HepG2 and 3T3-L1 cell lines. Two complementary assays were applied: one based on the quantification of adenylate kinase release as an indicator of cell membrane damage, and another relying on resazurin reduction to assess cellular metabolic activity. These complementary assays enable sensitive and reliable detection of cytotoxic effects across diverse cell types. This preliminary assessment was carried out to ensure that subsequent functional analyses would not be confounded by nonspecific cytotoxic effects. The results enabled the identification of non-cytotoxic concentration ranges (data about quantification of membrane damage are shown in Appendix A, and data about metabolic viability are shown in Appendix A), with 10 µM determined as the highest concentration that did not induce significant cytotoxicity and was therefore selected for subsequent experiments.

#### 2.4.3. Influence of JP-14 on the Live Organism in the *Danio rerio* Model, Preliminary Toxicity Studies

The survival and type of disorders occurring in *Danio rerio* larvae after 24 h incubation with the tested compound were assessed in order to find a concentration that does not visible negatively affect these organisms. All larvae from the Casper line exposed to a concentration of 50 µg/mL of JP-14 did not survive but no malformations were observed. Approximately 85% of larvae from the AB/TL line exposed to a concentration of 50 µg/mL of JP-14 developed disorders, which included cardiac edema, yolk edema or yolk deformation, and tail bend. Figure 14 shows a quantitative analysis of disorders in this group and representative pictures of larvae from each group. Casper larvae were selected for the study because the next step was to determine the effect of a safe concentration of JP-14 for larvae on lipid utilization or disorders after induction by fructose, and because Casper are transparent, they are suitable for methods where dyes are used. Preliminary toxicity studies were also performed on ABTL larvae because, in these larvae, such disorders can be observed that cannot be seen in Casper larvae, i.e., pigmentation disorders. However, no effect on pigmentation was observed after incubation of JP-14 with larvae. Our studies have shown that JP-14 may significantly affect the circulatory system at very high concentrations, and future safety studies related to this system will be needed. Finally, 10 µg/mL was chosen as a concentration suitable for testing biological effects while not causing any visible disturbances.

#### 2.4.4. Influence of JP-14 on Platelet Aggregation

It is known that α_2_-adrenergic receptors play a significant modulatory role in platelet aggregation. Epinephrine itself is a weak platelet agonist but notably enhances aggregation when combined with other stimuli like ADP or TXA_2_, acting mainly via the α_2A_-adrenergic receptors [56]. Moreover, TAAR1 have been discovered in platelets [57]; the potential impact of JP-14 on platelet function was subsequently evaluated.

The effect of JP-14, guanabenz, and RTI-7470-44 on platelet aggregation was investigated in rat whole blood using impedance aggregometry, a method that allows for evaluation under conditions that closely mimic physiological environments [58,59]. Platelet aggregation was induced by collagen, a key physiological agonist of platelet activation, whose use in ex vivo and in vitro models enables the assessment of platelet function under conditions that reflect physiological thrombus formation [60]. Collagen stimulation leads to a rise in intracellular calcium levels, triggering a series of downstream responses including platelet shape change, exposure of procoagulant phospholipids, secretion of ADP and serotonin (5-HT), and synthesis of thromboxane A_2_ (TXA_2_). These cascades ultimately result in the activation of the glycoprotein IIb/IIIa (GPIIb/IIIa) receptor complex, exposure of the sites that connect to fibrinogen, and platelet aggregation [61,62]. As a result, agents that interfere with any of these pathways may modulate the collagen-induced aggregation response [63]. On the other hand, multiple platelet agonists exert synergistic effects on aggregation by concurrently activating diverse receptors and signaling cascades, thereby amplifying the platelet activation response [64].

In our study, guanabenz was found to enhance collagen-induced platelet aggregation, increasing aggregation by 31.81% at 10 µM (*p* < 0.001) and by 22.73% at 100 µM (*p* < 0.05) (Figure 15a). Activation of α_2_-adrenergic receptors reduces intracellular cAMP through coupling to the Gi/z protein, thereby reducing a key inhibitory signal and augmenting Ca^2+^ increase induced by other agonists [65,66]. This may explain the facilitating effect of guanabenz, a potent α_2_-adrenergic receptor agonist on collagen induced platelet aggregation in vitro. As opposed to to guanabenz, compound JP-14 did not influence collagen induced platelet aggregation in a wide range of concentration (50–200 µM), (Figure 15). This effect may be attributed to the substantially lower affinity of JP-14 for the α_2A_-adrenoceptor compared to guanabenz (Ki = 126 nM vs. 6 nM, respectively).

In addition, we tested the effect of compound RTI-7470-44 on platelet aggregation induced by collagen. RTI-7470-44 at the highest concentration tested (100 µM) reduced platelet aggregation by 41.75% (*p* < 0.01) (Figure 15c); however, the underlying mechanism remains to be elucidated.

The compound JP-14 is not only a TAAR1 agonist but also a partial agonist of α_2_A-adrenergic receptors. Therefore, the lack of determination of its effect on insulin and glucose levels, for example, in a cellular model, is a limitation of these studies. The partial agonist activity of JP-14 at the α2A-adrenoceptor (~15%) may indicate a potential role in metabolic regulation, although its involvement has not yet been conclusively established. The assessment of JP-14-mediated effects on insulin secretion in cell models with high α2A-adrenoceptor expression represents a relevant direction for future investigations, and we plan to incorporate such analyses in our subsequent studies to clarify the mechanistic basis and possible metabolic consequences of JP-14 activity. Next, due to the beneficial metabolic properties of JP-14 presented in this manuscript, we will further investigate this compound’s effects on body weight and metabolic disorders, for example, in a mouse obesity model. Furthermore, given the receptor profile of JP-14, it is worthwhile to examine its effect on heart rate and blood pressure in the future, thus expanding the safety research on this compound. Furthermore, to ensure that the compound acts only peripherally (does not cross the BBB), as indicated by in silico data, we plan to conduct comprehensive studies on JP-14’s ability to penetrate the barrier cells, including its impact on its integrity.

So far, there are few studies described on the effect of TAAR1 agonists on body weight and metabolic disorders, but those that can be reviewed indicate that such compounds may prove to be a golden shot in the fight against this type of disorders [2,67,68]; therefore, we emphasize that it is worth developing research on JP-14 and other compounds whose mechanism of action is based on the effect on TAAR1.

## 3. Materials and Methods

### 3.1. Molecular Modeling

The following computational tasks were performed using Small-Molecule Drug Discovery Suite (Schrödinger Release 2025-1, Schrödinger, LLC, New York, NY, USA, 2025). The structures of JP-14, guanabenz, and 4-hydroxyguanabenz were prepared using LigPrep tool [69] with default parameters. As the JP-14 structure features a double bond, the compound exists in both E and Z configurations, both of which were examined in the present work.

Preparation of models of the biological targets was about optimizing the available 17 PDB structures of the TAAR1 using Protein Preparation Wizard [70]. Ligand docking was performed using the Standard Precision (SP) protocol of Glide [71], with a hydrogen bond interaction with Asp3.32 applied as a constraint to reflect the characteristic interaction of aminergic GPCRs.

Additionally, quantum mechanical calculations were carried out using Jaguar [72] to predict the lowest-energy conformers of the E and Z isomers of JP-14. These calculations were conducted in water as the solvent, using default settings and generating up to 5 conformers per isomer.

### 3.2. Chemistry

#### 3.2.1. Chemical and Analytical Methods

Reagents were purchased from commercial sources and used without further puri-fication. Thin-layer chromatography (TLC) was used to monitor reaction progress. TLC was carried out on chromatographic plates: Kieselgel 60 F254 (Merck, Darmstadt, Ger-many) visualized under UV light and with ninhydrin, vanillin solution, and cerium (IV) sulfate. Reaction monitoring and purity control were achieved by HPLC (Waters Alliance e2695, PDA detector, 200–800 nm; Waters Corporation, MA, USA) on a Chromolith SpeedROD RP-18e column (4.6 × 50 mm), using a gradient of acetonitrile/0.1% TFA (0–100%) at 5 mL/min with a 3 min run time.” The identity and purity of the final compounds were confirmed by ultra-performance liquid chromatography–mass spectrometry (UPLC-TQD MS; Waters Corporation, MA, USA). Chromatographic separation employed a BEH C18 column (2.1 × 100 mm, 1.7 µm) equipped with a VanGuard precolumn (2.1 × 5 mm, 1.7 µm). A linear gradient from 95% to 0% of eluent A (0.1% formic acid in water) was applied over 10 min, followed by 2 min of isocratic elution with 100% eluent B (0.1% HCOOH in MeCN), with the column maintained at 40 °C and the flow rate set to 0.3 mL/min. The UPLC–MS analysis confirmed that the purity of all final compounds exceeded 95%. NMR analyses (1H, 13C, and 19F) were performed on a 500 MHz FT-NMR spectrometer (JEOL JNM ECZR500 RS1, Tokyo, Japan) in DMSO-d6 or CD3OD using TMS as the internal standard. J values are given in Hertz (Hz).

#### 3.2.2. Synthetic Procedures

A detailed description of the synthesis of the most promising compound, JP-14, is provided below, while comprehensive characterization data—including ^1^H, ^13^C, and ^19^F NMR spectra and LC/MS chromatograms—for JP-14 and the remaining compounds are included in the Appendix A.

##### (*E*)-2-((3,5-Dichloropyridin-4-yl)Methylene)Hydrazine-1-Carboximidamide (JP-14; 10)

The 3,5-dichloroisonicotinaldehyde (1 equiv., 200 mg, 1.14 mmol) and aminoguanidine hydrochloride (1.1 equiv., 139 mg, 1.25 mmol) were mixed with methanol (5 mL) and stirred for 2–3 h at reflux. The obtained reaction mixture was then cooled to 25 ± 5 °C, and diethyl ether (Et_2_O) and petroleum ether (PE) were added. The pure product was then crystallized from the reaction mixture. The obtained precipitate was washed with a Et_2_O and PE (1:1 (*v*/*v*)) and then dried under reduced pressure. Yield: 49.5%, beige powder, ^1^H NMR (500 MHz, DMSO-d_6_): δ 8.69 (s, 2H), 8.41 (s, 1H), 7.84 (br s, 4H). ^13^C NMR (126 MHz, DMSO-d_6_): δ 155.89, 148.93 (2C), 140.83, 136.92, 131.01 (2C). Formula: C_7_H_7_Cl_2_N_5_; MS (ESI^+^) m/z: 233.83 [M + H^+^].

### 3.3. In Vitro Assays

#### 3.3.1. Cell Culture

##### T3-L1 Adipocytes

The 3T3-L1 mouse embryonic fibroblast cell line (ATCC CRL-11605) was cultured according to the standard protocol in Dulbecco’s Modified Eagle Medium (DMEM) supplemented with 10% bovine calf serum, 100 µg/mL streptomycin, and 100 IU/mL penicillin, at 37 °C in a humidified atmosphere containing 5% CO_2_. Cells were seeded at a density of 10,000 cells/well in collagen type I–coated (10 µg/mL) 96-well plates and allowed to attach for 24 h. Differentiation was initiated by replacing the medium with induction medium composed of DMEM, 0.5 mM 3-isobutyl-1-methylxanthine (IBMX), 1 µM dexamethasone, and 10% fetal bovine serum (FBS). After 48 h (differentiation day 2), the medium was replaced with DMEM supplemented with 10% FBS and 10 µg/mL human recombinant insulin. After an additional 24 h, the medium was replaced with DMEM supplemented with 10% bovine calf serum, 100 µg/mL streptomycin, and 100 IU/mL penicillin. The medium was refreshed every 24 h for 10 consecutive days to allow full differentiation into adipocytes.

##### HepG2 Hepatocytes

The HepG2 human hepatocellular carcinoma cell line (ATCC^®^ HB-8065™) was cultured in DMEM supplemented with 10% FBS, 100 µg/mL streptomycin, and 100 IU/mL penicillin at 37 °C in 5% CO_2_. Cells were seeded in 96-well plates at a density of 10,000 cells/well and used for experiments 24 h after plating.

##### Compound Preparation and Treatment

Stock solutions of test compounds were prepared at a concentration of 10 mM in dimethyl sulfoxide (DMSO). At least 1 mg of each substance was carefully measured and mixed with the right amount of DMSO. A series of dilutions were made in DMSO and then further diluted in phosphate-buffered saline before being introduced into the cell culture medium. Precipitation or opalescence was assessed prior to application. Notably, many compounds precipitated at concentrations of 100 µM in PBS.

#### 3.3.2. Cytotoxicity and Viability Assays

##### Membrane Integrity Assay

The damage of cell membrane was evaluated with the ToxiLight BioAssay Kit from Lonza (Visp, Switzerland). This test is very sensitive and measures cell damage by looking at the release of adenylate kinase (AK). After 24 h of treatment, 5 µL of culture supernatant was transferred to a 384-well white plate (PerkinElmer, Singapore), followed by the addition of 20 µL of Adenylate Kinase Detection Reagent (AKDR). After a 10 min incubation at room temperature, luminescence was measured using the POLARstar Omega plate reader (BMG Labtech, Ortenberg, Germany). Results were expressed as a percentage relative to a lysed control (100% cell death).

##### Cell Viability Assay

Cell viability was measured using PrestoBlue reagent (Thermo Fisher Scientific, Waltham, MA, USA), following the manufacturer’s instructions. After 24 h of compound treatment, PrestoBlue reagent was added to each well at 10% of the remaining medium volume. The plate was incubated at 37 °C for 15 min, and fluorescence was measured using the POLARstar Omega plate reader (excitation 530 nm, emission 580 nm). Results were expressed as a percentage of viable cells relative to the DMSO-treated control.

#### 3.3.3. Lipid Accumulation and Glucose Consumption

##### AdipoRed Assay

On day 10 of differentiation, cells were treated with test compounds for 24 h. To investigate the involvement of the TAAR1 receptor, cells were pre-treated for 15 min with the selective TAAR1 antagonist RTI-7470-44 (40 nM) before compound exposure. After incubation, the culture medium was removed, and wells were gently rinsed with 200 µL PBS and refilled with 200 µL PBS. AdipoRed reagent (5 µL; Lonza) was added to each well, mixed, and incubated at room temperature for 10 min. Fluorescence was measured using the POLARstar Omega plate reader (excitation 485 nm, emission 535 nm).

##### Glucose Consumption (Amplex Red Assay)

The glucose uptake was measured with the Amplex^®^ Red Glucose/Glucose Oxidase Assay Kit from Thermo Fisher Scientific, following the guidelines provided by the maker. Cell culture supernatants were collected and incubated with a reaction mix containing 100 μM Amplex^®^ Red reagent, 0.2 U/mL horseradish peroxidase (HRP), and 1 U/mL glucose oxidase at room temperature for 30 min in the dark. Fluorescence was measured at 530 nm excitation and 590 nm emission using the POLARstar Omega plate reader (BMG Labtech).

##### High Content Analysis of Lipid Accumulation

Lipid accumulation and phospholipidosis were evaluated using the HCS LipidTOX Phospholipidosis and Steatosis Detection Kit (Thermo Fisher Scientific). Cells were cultured in black, clear-bottom 96-well plates and treated as described. Next, cells were fixed with 4% paraformaldehyde (15 min), washed with PBS, and stained with LipidTOX Green Neutral Lipid Stain (for steatosis) and LipidTOX Red Phospholipidosis Detection Reagent, prepared according to the manufacturer’s instructions. Nuclei were counterstained with Hoechst 33342 (2 µg/mL). Image acquisition and quantitative analysis were performed using High Content Screening (HCS) technology on the ImageXpress XLS system (Molecular Devices). Lipid droplet and phospholipid content were quantified using automated image analysis pipelines.

#### 3.3.4. Receptor-Binding Assay

The competitive radioligand-binding assay assessed affinity for α_1_- and α_2_-adrenoceptors. Experiments were carried out in the rat cerebral cortex. The radioligands [3H]prazosin (84.2 Ci/mmol, α1-adrenoceptor; PerkinElmer, MA, USA, cat. no. NET823001MC) and [3H]clonidine (70.5 Ci/mmol, α_2_-adrenoceptor; PerkinElmer, Waltham, MA, USA, cat. no. NET613250UC) were utilized. The membrane preparation and assay technique followed the previously reported protocol [10]. Radioactivity was measured using the MicroBeta2 scintillation counter (PerkinElmer, Kraków, Poland). The data was adjusted to match a single-site curve-fitting formula using Prism 8 (GraphPad Software, La Jolla, CA, USA), and the Ki values were estimated by means of the Cheng–Prusoff formula.

#### 3.3.5. Intrinsic Activity Assay

Intrinsic α_2A_-adrenoceptor activity was assessed using Tango assay technology, which utilizes a mammalian-optimized beta-lactamase (bla) reporter gene in conjunction with a FRET-enabled substrate to provide reliable and sensitive detection in cells. The intrinsic activity test was conducted based on the guidelines provided by the maker of the test kit (Invitrogen, Life Technologies, Carlsbad, CA, USA). The cells were harvested and resuspended in Assay Medium to a density of 312,500 cells/mL. A quantity of 32 μL per well of the cell suspension was added to the Test Compound wells, the Unstimulated Control wells, and Stimulated Control wells and incubated for 16–24 h. To perform the agonist assay, 8 μL of a 5-fold higher concentration of agonist in assay medium was added to the cells. To perform the antagonist assay, 4 μL of a 10-fold higher concentration of agonist in assay medium was added to the cells, and after 30 min of incubation, 4 μL of a 10-fold higher concentration of a standard agonist with an EC_80_ (brimonidine 10^−7^ M) in assay medium was added. The agonist and antagonist plates were then incubated in a humidified incubator at 37 °C and 5% CO_2_ for 5 h. After incubation, cells were loaded with 8 μL of LiveBLAzer™-FRET B/G substrate mix (CCF4-AM), covered for light protection, and incubated at room temperature for 2 h.

The intrinsic activity at the α_2B_-adrenoceptor and 5-HT_2C_ serotoninergic receptor will be assessed by luminescence detection of calcium mobilization using the recombinant expressed jellyfish photoprotein aequorin on cells with stable expression of the relevant receptors. The determination of intrinsic activity will be performed in accordance with the manufacturer’s Aequoscreen methodology—Perkin Elmer.

Measurements were performed with adrenergic α_2B_- or 5-HT_2C_ AequoScreen cell line (PekinElmer cat. no.ES-031-AF/ES-318-AF). The cell density in the 96-well format was 5000 cells per well. Cell preparation for the assay was performed according to the manufacturer’s instructions for AequoScreen (PerkinElmer, Waltham, MA, USA). A series of compound concentrations (50 μL/well) was diluted in 0.1% BSA (Intergen, cat. no. 3440-75) with assay buffer (DMEM/F-12, Invitrogen, cat. no. 11039) and prepared in white ½ Area Plate—96-well microplates (PerkinElmer, cat. no. 6005560). The cells suspension was applied onto the ligands with the injectors of the POLARstar optima reader (BMG Labtech, Ortenberg, Germany). Detection of agonist: Oxymetazoline (Sigma, Cat. no. O2378) was used as an agonist for the α_2B_-adrenoceptor. The concentration and dilution series having four replicates were prepared as instructed in the AequoScreen Starter Kit Manual (PerkinElmer, Waltham, MA, USA). The wells received cell injections along with either the standard agonist or the compounds being evaluated, while the POLARstar optima reader documented the emitted light over a duration of 20 s. In the antagonist evaluation, cells were administered (50 μL) into the assay plate alongside antagonist solutions (50 μL). A single concentration of agonist (oxymetazoline) was injected (50 μL, final concentration EC_80_) into the preincubated (15–20 min) plate with cells and antagonist, and the emitted light was recorded for 20 s.

The intrinsic activity of the TAAR1 receptor will be assessed according to the manufacturer’s protocol, using the Hit Hunter^®^ cAMP assay by DiscoverX. This assay enables monitoring of GPCR activation in cells stably expressing TAAR1 by measuring changes in the level of the secondary messenger, cAMP. The cAMP Hunter cell lines were cultured from frozen stocks following standard procedures. Cells were plated in white-walled 384-well microplates at a total volume of 20 µL per well and incubated at 37 °C for the required period before analysis. To assess agonist activity, cells were exposed to the test compound to trigger a response. The culture medium was then removed and replaced with 15 µL of a 2:1 mixture of HBSS and 10 mM Hepes containing the cAMP XS+ antibody reagent. Sample stocks were diluted to prepare a 4× concentration in assay buffer. Subsequently, 5 µL of the 4× sample was added to each well, and the plates were incubated at either 37 °C or room temperature for 30 or 60 min. The vehicle was used at a concentration of 1%. After incubating the compounds for the required time, the assay signal was produced by adding 20 µL of the cAMP XS+ ED/CL lysis cocktail and incubating for one hour. This was followed by a three-hour incubation with 20 µL of the cAMP XS+EA reagent at room temperature. After signal development, the microplates were read using a PerkinElmer Envision Reader to detect chemiluminescence.

#### 3.3.6. In Vitro Platelet Aggregation Tests

In vitro platelet aggregation assays were conducted using freshly collected whole blood from rats, utilizing the Multiplate platelet function analyzer (Roche Diagnostics, Mannheim, Germany)—a five-channel device that measures electrical impedance, following methods described previously [73]. Upon stimulation, platelets adhere to and aggregate on dual metal sensors within the test cuvette, causing an increase in electrical resistance, which corresponds to the number of platelets attached to the electrodes.

Blood was collected from the carotid artery of the rat into tubes coated with hirudin (S-Monovette, Sarstedt, Germany). A volume of 300 μL of the anticoagulated blood was mixed with 300 μL of prewarmed isotonic saline that contained the test compound or a vehicle control (0.1% DMSO) and incubated at 37 °C for 3 min under continuous stirring. Aggregation was triggered by adding collagen (final concentration: 1.6 µg/mL). Platelet activity was monitored for 6 min. Multiplate V2.04 software calculated the area under the curve (AUC) for each measurement to determine average platelet response.

The research has been reported to and approved by the Animal Welfare Committee of the Faculty of Pharmacy of Jagiellonian University Medical College (resolution no 1/2024, 15 January 2024).

### 3.4. In Vivo Assays

#### 3.4.1. Inhibition of Gastric Emptying (Mouse Model)

Mice were acquired from the Jagiellonian University Animal House located at the Faculty of Pharmacy in Krakow, Poland. The study was carried out in accordance with the guidelines outlined in the Declaration of Helsinki and received approval from the Local Ethics Committee for Animal Experiments in Krakow, Poland (Permissions No: 797/20023, dated 14 December 2023, and No: 9A/205, dated 27 March 2025).

Fifty male C57BL/6 mice (22–27 g) were housed in standard cages while maintaining a 12:12-h light/dark cycle, with food and water available ad libitum. The experimental procedures were performed from 10:00 a.m. to 14:00 p.m.

The experiment was conducted according to Miyasaka et al. [74] and Kotańska et al. [18], with some minor modifications. Mice were subjected to a 20 h food deprivation and a 2 h water deprivation prior to testing. The animals received a vehicle (0.25 mL/mouse) alone or along with various compounds: JP-14 (10 mg/kg b. w.), RT (5 mg/kg body weight), metformin (100 mg/kg b. w.), or loperamide (reference compound; 10 mg/kg b. w.) via gastric gavage. Following a 45 min interval, phenol red (0.5 mL, 0.05% *w*/*v*) combined with glucose (1 g/kg body weight) was administered through the same route. Additionally, each group received an intraperitoneal injection of heparin (2500 U/mouse) 20 min prior to the collection of stomach contents. Seven mice treated with the vehicle were sacrificed immediately after the phenol red administration to serve as a standard control (representing 100% phenol red in the stomach at 0 min), establishing a baseline for maximum absorbance (100%). The leftover mice were sacrificed 30 min later. The stomachs were removed and homogenized using an OMNI homogenizer (OMNI International Company, Kennesaw, GA, USA), with 10 mL of 0.1 N NaOH. To 5 mL of homogenate, 0.5 mL of 20% *w*/*v* solution of trichloroacetic acid was incorporated to precipitate the proteins. Subsequently, the samples were centrifuged at 600× *g* for 15 min at a temperature of 20 °C, and 0.25 mL of supernatant was mixed with 1 mL of 0.5 N NaOH to achieve maximum color intensity. The concentration of phenol red was measured using spectrophotometry, with absorbance evaluated at a wavelength of 560 nm (MuLTISKAN.GO, Thermo Fisher Scientific Inc., Waltham, MA USA). For each mouse, the percentage of gastric emptying (GE %) was calculated using the following equation:(1)$$GE\%=\frac{\left(Ac0-Ab\right)}{\left(Ac0-Ac30\right)}\times 100\%$$
where: Ac0 is the absorbance of the control gastric sample collected immediately after dye administration, and Ac30 is the absorbance of the control gastric sample collected 30 min after dye administration, i.e., Ac0–Ac30 = 100% gastric emptying.

#### 3.4.2. In Vivo Toxicity Assay

The embryos of zebrafish were obtained through the natural reproduction of adult specimens (specifically line ABTL and Casper), which were housed in a continuously recirculating closed-system aquarium operating on a light/dark cycle of 14 h of light and 10 h of dark at a temperature of 28 degrees Celsius. The breeding process took place at the Zebrafish Core Facility at Jagiellonian University (JU ZCF), located within the Institute of Zoology and Biomedical Research, Department of Evolutionary Immunology, based in Kraków. JU ZCF is an authorized breeding and research establishment, as registered with the District Veterinary Inspectorate in Krakow and the Ministry of Science and Higher Education, having record numbers 022 and 0057, respectively. All conducted experiments complied with the European Community Council Directive 2010/63/EU regarding the treatment and use of laboratory animals, which was finalized on 22 September 2010 (Chapter 1, Article 1 no.3), along with the National Journal of Law act from 15 January 2015 concerning the Protection of Animals Used for Scientific or Educational Purposes (Chapter 1, Article 2 no.1). The handling of all animals was done in strict accordance with established good practice guidelines for animal welfare created by relevant national and/or local animal welfare organizations. In accordance with the European Directive 2010/63/EU and Polish legal regulations (O.J. of 2015, item 266), all procedures conducted in this study involving zebrafish and euthanasia are exempt from the requirement for Ethics Committee approval.

To determine the toxicity of JP-14 to zebrafish larvae, 76 h post-fertilized (hpf) larvae (900 pieces, ten per well) were kept in a 6-well plate and exposed to three concentrations of compound JP-14 diluted with embryonic medium (E3), whereas untreated larvae served as control (E3 alone). Each concentration was incubated in twelve wells. The mortality rate and disturbances were determined using a microscope (OPTA-TECH MN16, Warszawa, Poland) after 24 h of exposure.

#### 3.4.3. Nile Red Fluorescence Fat Metabolism Assay

The study was conducted based on the method described earlier by Jones et al. [29]. Starting at 3 days after fertilization (dpf), larvae were placed into 6-well plates and incubated with the test solutions (compounds in E3: JP-14 at a concentration of 10 μg/mL or resveratrol or E3 alone) for 48 h. Resveratrol solution (20 μg/mL) was used as the reference compound. After 24 h, the solutions were replaced with new ones containing Nile red dye (9-diethylamino-5H-benzo[α]phenoxazine-5-one, Pol-aura, Poland) at a concentration of 100 ng/mL. Nile red is a selective fluorescent lipid dye used to stain lipid droplets. Incubation with the dye was therefore 24 h. Stock solution of Nile red was prepared earlier by dissolving it in acetone to a concentration of 500 μg/mL. After the incubation period, the larvae were rinsed 3 times with 0.1 M phosphate-buffered, pH 7.4 (PBS, Sigma-Aldrich, Schnelldorf, Germany) and anesthetized with cold 0.1 mg tricaine solution (Sigma-Aldrich, Schnelldorf, Germany). Next, they were visualized under a fluorescence microscope (Zeiss.Discovery.V8, Zeiss, Oberkochen, Germany) and analyzed with ImageJ FIJI program (version 2.17.0) [75].

#### 3.4.4. Measurement of the Amount of Neutral Lipids in Zebrafish Larvae After Induction Metabolic Disorders with Fructose

Zebrafish larvae 4 dpf were incubated for another 24 h with 4% fructose (final solution) in embryo medium to induce metabolic disorders. Tested solutions (compounds in E3: JP-14 at concentration of 20 μg/mL or resveratrol of 40 μg/mL) were added to fructose solution (8% in E3). The final concentration of JP-14 was 10 μg/mL or resveratrol of 20 μg/mL. The control group was incubated with E3 only. Incubation was carried out in a 6-well plate (10 larvae/well) to which 3 mL of liquid was added per well. Larvae were then collected, euthanized with tricaine, and transferred to 4% formaldehyde solution. After 24 h, larvae were washed 3 times with PBS (Sigma-Aldrich, Germany) and stained for neutral lipids, similar to previous studies [34]. Larvae incubated for 1 h in a solution of 0.25% oil red O (ORO, Merck, Darmstadt, Germany) in 60% isopropanol (Merck, Germany). The samples were then rinsed three times with 60% isopropanol and washed 6 times with PBS, then imaged on a microscope (OPTA-TECH MN16, Warszawa, Poland) and analyzed with the ImageJ FIJI program [75]. Next, the larvae were transferred to tubes (10 larvae/tube), and 0.2% Trixon-X 100 solution (Merck, Germany) in PBS was added and ground using a homogenizer (OMNI homogenizer, OMNI International Company, USA). The homogenate was left for 2 h, during which time each test tube was mixed every 10 min on a vortex for 1 min. Next, the samples were centrifuged (10 min, 3000 rpm, MPW-260R, MPW Med Instruments, Warszawa, Poland), and the amount of dye in supernatant was determined spectrophotometrically (MuLTISKAN.GO, Thermo Fisher Scientific Inc. Waltham, MA, USA) at a wavelength of 495 nm.

### 3.5. In Silico Prediction of Chemical, Pharmacokinetic, and Drug-Likeness

The free SwissADME web tool (http://www.swissadme.ch, accessed on 20 April 2025) provided the parameters, including physicochemical properties (molecular weight, numbers of rotatable bonds, numbers of hydrogen bond donors, numbers of hydrogen bond acceptors, and fraction Csp3), lipophilicity, water solubility, gastrointestinal absorption, blood–brain barrier permeant, P-glycoprotein substrate, drug-likeness rules, and pan assay interference compounds methods. Absorption, distribution, metabolism, excretion/elimination, and toxicity (ADMET) were calculated using SwissADME [43]. The free pkCSM (http://biosig.unimelb.edu.au/pkcsm/prediction, accessed on 20 April 2025) web tool was used to predict the toxicity of JP-14 [44]. The results for JP-14 were compared with results for guanabenz.

### 3.6. Statistical Analysis

The statistical calculations were conducted using the GraphPad Prism 7.0 application (GraphPad Software, La Jolla, CA, USA). The Shapiro–Wilk test was used for analyzing the normal distribution of the data. For the statistical significance of the findings, a nonparametric Kruskal–Wallis test with Dunn’s post hoc or parametric one-way ANOVA with Tukey’s post hoc (for data with normal distribution) tests were used. The data are displayed as the mean ± standard deviation (SD) or median. Statistical significance was determined for variations when the significance level was below 0.05 (*p* < 0.05).

## 4. Conclusions

The aminoguanidine derivative JP-14 emerged from a focused medicinal chemistry campaign as a novel, functionally active TAAR1 agonist (EC_50_ = 11.29 ± 1.4 µM), exhibiting negligible activity at the 5-HT_2C_ and α_1_-adrenergic receptors, and only weak submicromolar interactions with α_2_-adrenergic receptor subtypes. In HepG2 cells, JP-14 significantly enhanced glucose consumption (~32%), an effect comparable to that of metformin and tyramine, and abolished upon co-treatment with the selective TAAR1 antagonist RTI-7470-44, confirming TAAR1-mediated activity. In adipocyte-based assays, JP-14 reduced both neutral lipid accumulation and phospholipidosis during 3T3-L1 differentiation, with these anti-lipogenic effects also fully reversed by TAAR1 blockade. In vivo studies in zebrafish larvae demonstrated that JP-14, at a non-toxic concentration of 10 µg/mL, promoted yolk-sac lipid utilization and prevented fructose-induced yolk enlargement, supporting its efficacy in modulating lipid metabolism in a whole-organism context. Furthermore, intragastric administration of JP-14 (10 mg/kg) in mice resulted in a robust inhibition (~66%) of gastric emptying, comparable to loperamide. This effect was partially attenuated (~25% inhibition) in the presence of RTI-7470-44, suggesting a predominantly TAAR1-dependent mechanism, potentially influenced by pharmacokinetic differences between the agonist and antagonist. JP-14 did not alter collagen-induced platelet aggregation across a broad concentration range (50–200 µM), in contrast to guanabenz, which significantly enhanced aggregation—highlighting the weak α_2_A-adrenoceptor activity of JP-14 and its minimal impact on platelet function. Collectively, these data identify JP-14 as a promising lead compound with multimodal metabolic activity, capable of regulating hepatic glucose uptake, adipocyte lipid storage, systemic lipid utilization, and gastric motility through mechanisms substantially mediated by TAAR1. These findings support the further exploration of JP-14 as a candidate for therapeutic intervention in obesity and related metabolic disorders.

## Figures and Tables

**Figure 1 ijms-26-10033-f001:**
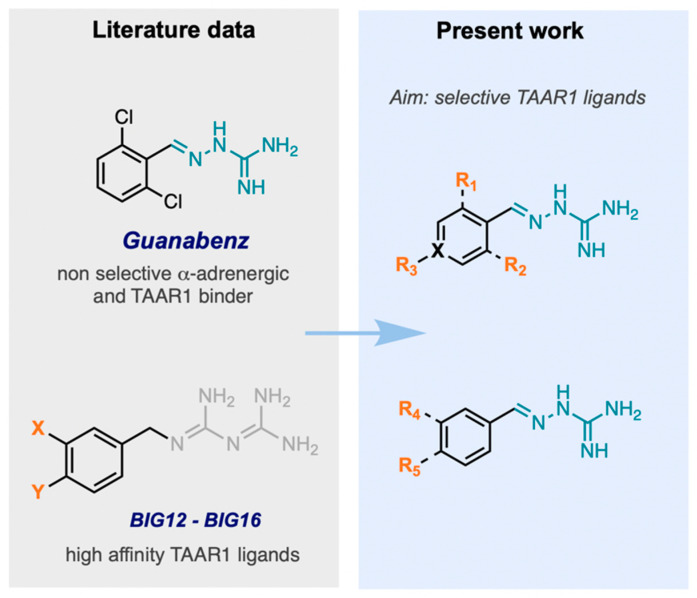
Design scheme of the novel TAAR1 series. X = C, N, R_1_, R_2_ = F, Cl, R_3_ = H, CN, R_4_ = OCH_3_, CN, R_5_ = OCH_3_, F, CN, (CH_3_)_2_N.

**Figure 2 ijms-26-10033-f002:**
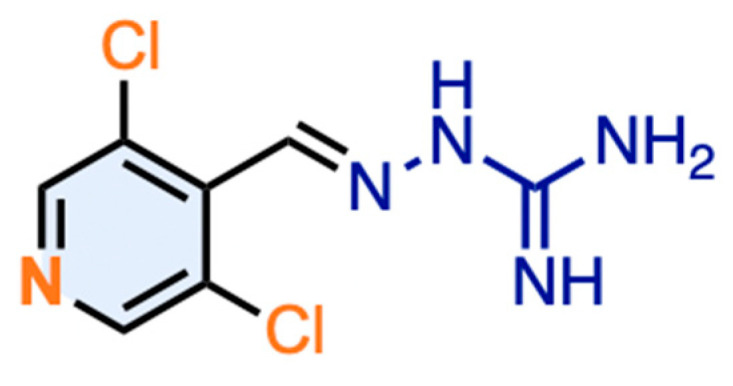
Chemical structure of selected lead compound JP-14.

**Figure 3 ijms-26-10033-f003:**
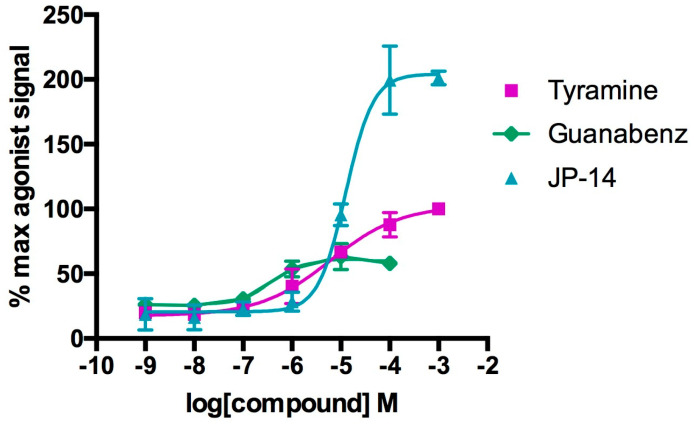
The intrinsic activity of the tested active compounds toward TAAR1 was evaluated based on their ability to activate the receptor in functional assays. The results were expressed as a percentage of the maximal response induced by the full agonist, tyramine. Data represent means ± SD from 2 to 4 independent experiments, each performed in duplicate.

**Figure 4 ijms-26-10033-f004:**
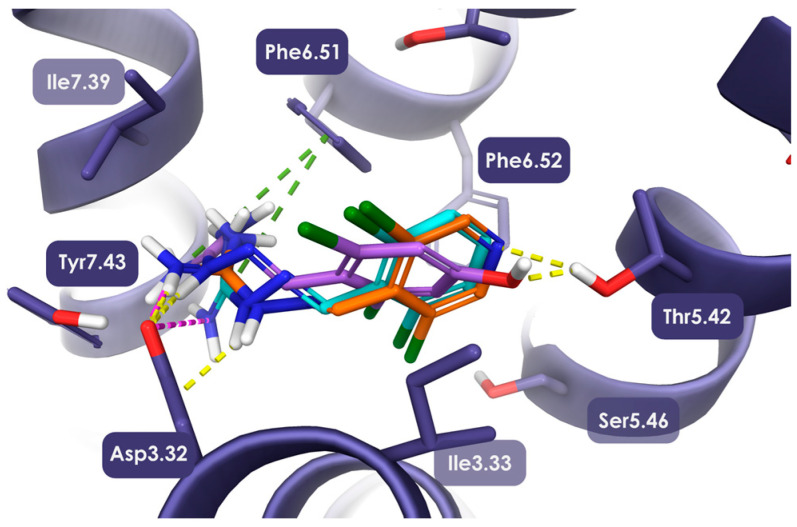
Binding mode of E-JP-14 (orange), guanabenz (cyan) and 4-OH-guanabenz (purple) in the 8ZSS structure of TAAR1. The guanidine group of E-JP-14 and the 2,6-dichloro-4-hydroxyphenyl moiety of guanabenz and its derivate formed a hydrogen bond and a salt bridge with Asp3.32 (yellow and pink dashed lines, respectively). Additionally, this group was stabilized by a π-cation interaction with Phe6.51 (green dashed line). Within the orthosteric pocket, the dichloropyrimidine moiety was stabilized by a hydrogen bond with Thr5.42.

**Figure 5 ijms-26-10033-f005:**
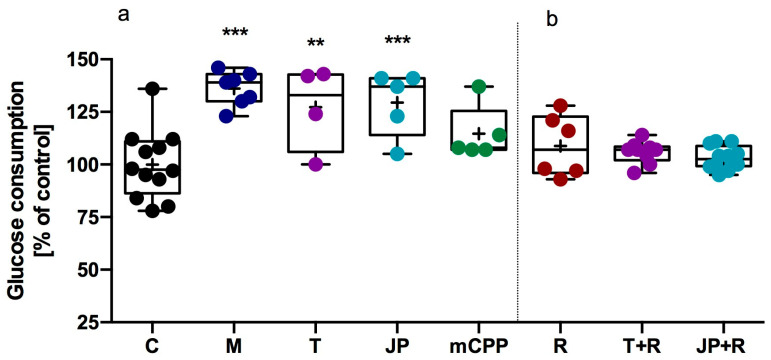
Effects of test compounds and TAAR1 antagonist on glucose consumption in HepG2 cells. Glucose consumption following treatment with test compounds (**a**). Glucose consumption in cells co-treated with the TAAR1 antagonist RTI-7470-44 (40 nM) to assess receptor involvement (**b**). C—control, M—metformin, T—tyramine, JP—JP-14, mCPP—meta-chlorophenylpiperazine, R—RTI-7470-44; mean (marked with “+”) ± SD (box); whiskers indicate minimum and maximum values; median (marked with a line); *n* = 5–12; one-way ANOVA followed by Dunnett’s post hoc test; ** *p* ≤ 0.01, *** *p* ≤ 0.001 vs. vehicle (control).

**Figure 6 ijms-26-10033-f006:**
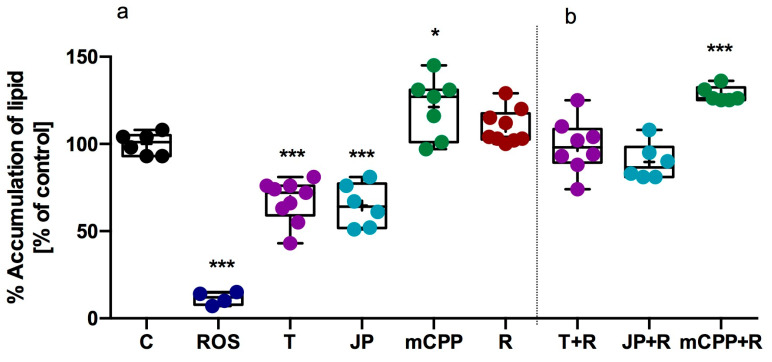
Effects of test compounds and TAAR1 antagonist on accumulation of lipid cells in 3T3-L1 cells. Accumulation of lipid following treatment with test compounds (**a**). Accumulation of lipid in cells co-treated with the TAAR1 antagonist RTI-7470-44 (40 nM) to assess receptor involvement (**b**). C—control, ROS—rosiglitazone, T—tyramine, JP—JP-14, mCPP—meta-chlorophenylpiperazine, R—RTI-7470-44; mean (marked with “+”) ± SD (box); whiskers indicate minimum and maximum values; median (marked with a line); *n* = 4–10; one-way ANOVA followed by Dunnett’s post hoc test; * *p* ≤ 0.05, *** *p* ≤ 0.001 vs. vehicle (control).

**Figure 7 ijms-26-10033-f007:**
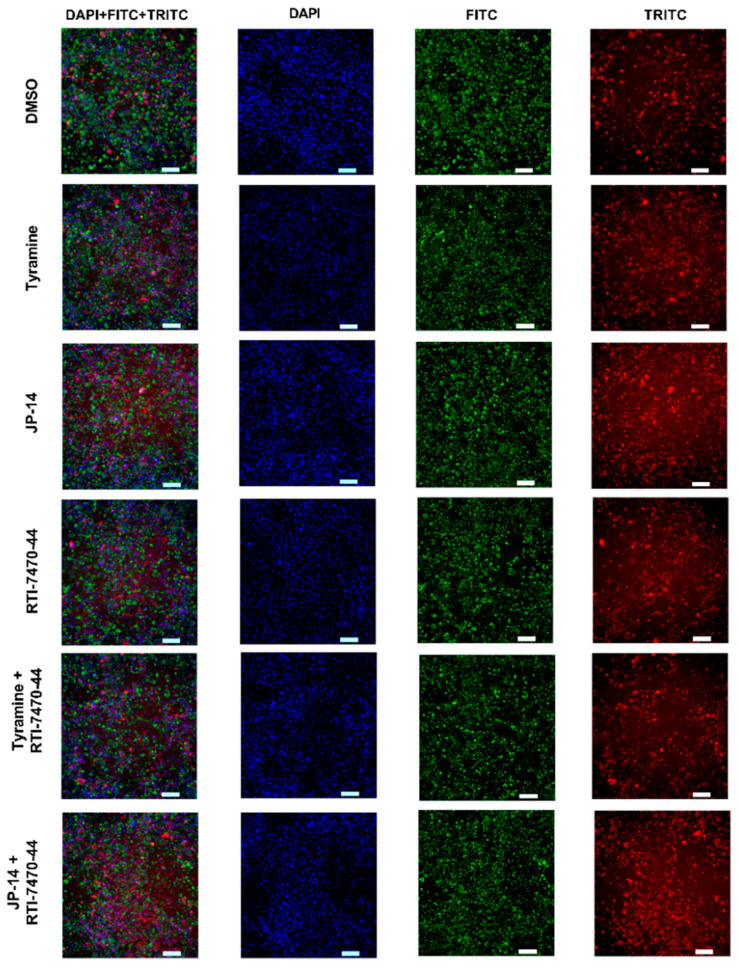
High-content screening images of phospholipidosis and steatosis in 3T3-L1 adipocyte cells. Images of 3T3-L1 adipocyte cells pretreated with control (0.5% DMSO), tyramine, RTI-7470-44, JP-14, and combinations of RTI-7470-44 with JP-14 or tyramine. Fluorescent staining was performed with the LipidTOX assay: Green (FITC channel—fluorescein isothiocyanate) labels lysosomal phospholipid accumulation; Red (TRITC channel—tetramethylrhodamine isothiocyanate) labels neutral lipid droplets indicative of steatosis; Blue (DAPI—4′,6-diamidino-2-phenylindole, Hoechst 33342) marks nuclei. Images were acquired using High Content Screening technology (ImageXpress XLS, Molecular Devices, San Jose, CA, USA). A scale bar representing 100 µm is shown in each image.

**Figure 8 ijms-26-10033-f008:**
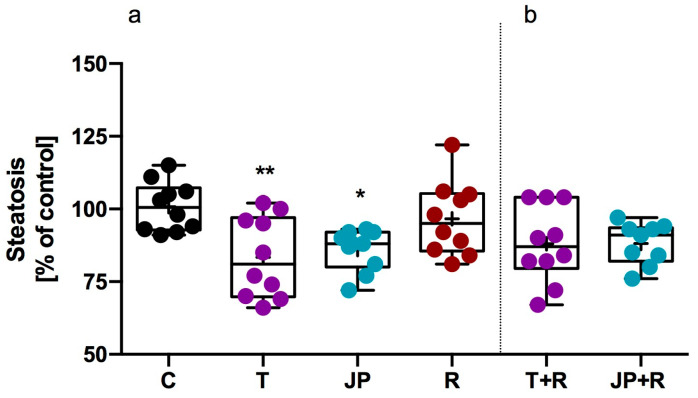
The effects of tested compounds on steatosis in 3T3-L1 cells. The impact of the compounds on steatosis (**a**). The effect of TAAR1 receptor blockade on steatosis following co-treatment with the TAAR1 antagonist RTI-7470-44 (**b**). C—control, T—tyramine, JP—JP-14, R—RTI-7470-44, T + R—tyramine + RTI-7470-44; mean (marked with “+”) ± SD (box); whiskers indicate minimum and maximum values; median (marked with a line); *n* = 9–12; one-way ANOVA followed by Dunnett’s post hoc test; * *p* ≤ 0.05, ** *p* ≤ 0.01 vs. vehicle (control).

**Figure 9 ijms-26-10033-f009:**
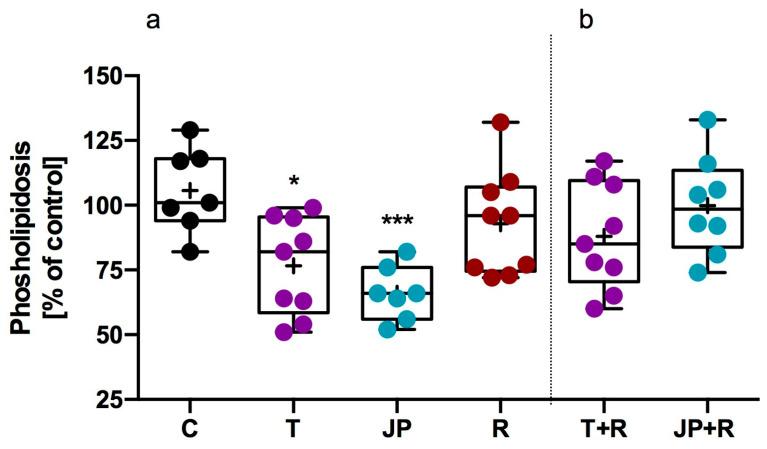
The effects of tested compounds on phospholipidosis in 3T3-L1 cells. The impact of the compounds on phospholipidosis (**a**). The effect of TAAR1 receptor blockade on phospholipidosis following co-treatment with the TAAR1 antagonist RTI-7470-44 (40 nM) (**b**). C—control, T—tyramine, JP—JP-14, R—RTI-7470-44; results are presented as percentages relative to vehicle-treated control cells; mean (marked with “+”) ± SD (box); whiskers indicate minimum and maximum values; median (marked with a line); *n* = 7–10. Statistical analysis was performed using one-way ANOVA followed by Dunnett’s post hoc test. Significance compared to control is indicated as follows: * *p* ≤ 0.05, *** *p* ≤ 0.001 vs. vehicle.

**Figure 10 ijms-26-10033-f010:**
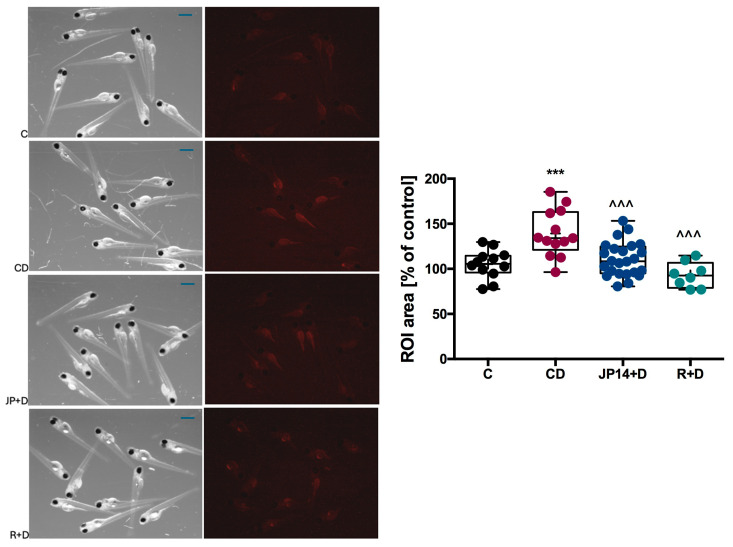
Lipid-reducing activity of JP-14 at concentration of 10 µg/mL in the Nile red fat metabolism assay. Representative pictures and quantitative image analysis of lipid in *Danio rerio* larvae after E3 alone, 10 µg/mL JP-14 or 20 µg/mL resveratrol exposure. C—control, CD—control dyed, JP14+D—JP-14 dyed, R+D—resveratrol dyed; mean (marked with “+”) ± SD (box); whiskers indicate minimum and maximum values; median (marked with a line); *n* = 8–24; one-way ANOVA test; Tuckey’s post hoc test; *—vs. control without staining (C), ^—vs. control dyed (E3 dyed group, CD); difference was considered when ***, ^^^ *p* < 0.001, magnification 2×, a scale bar representing 1000 µm is shown in each image.

**Figure 11 ijms-26-10033-f011:**
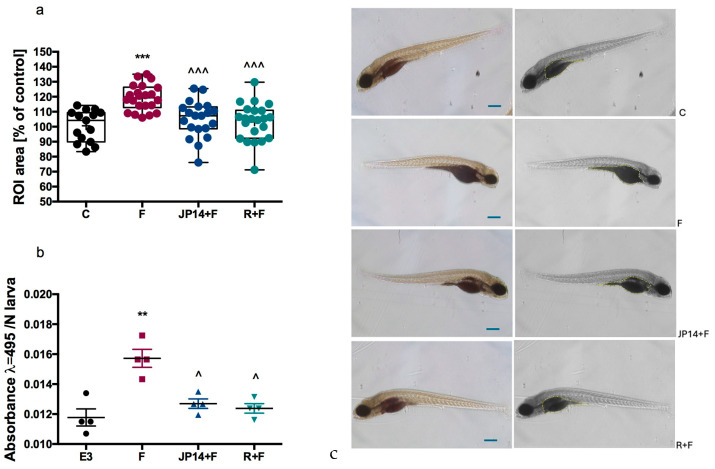
Oil Red O staining: quantitative results of size of the yolk—ROI area (**a**); the intensity of the larva’s color-dependent on the presence of lipids (**b**); representative pictures of *Danio rerio* larvae and ROI area (yellow line) after incubation with fructose and tested compounds, magnification: 4× (**c**); C—control larva—incubated with E3 alone, F—larva after incubation with 4% fructose solution, JP14+F—larva after incubation with 4% fructose solution and JP-14 at concentration 10 µg/mL, JP14+R—larva after incubation with 4% fructose solution and resveratrol at concentration 20 µg/mL; mean (marked with “+”) ± SD (box); whiskers indicate minimum and maximum values, median (marked with a line); *n* = 20–24 (**a**) or Mean ± SD, *n* = 4 (**b**); Kruskal–Wallis’s test; Dunn’s post hoc test (**b**); *—vs. C group (E3 alone), ^—vs. F group (E3 + fructose); difference was considered when ^ *p* < 0.05, ** *p*< 0.01, ***, ^^^ *p* < 0.001, a scale bar representing 500 µm is shown in each image.

**Figure 12 ijms-26-10033-f012:**
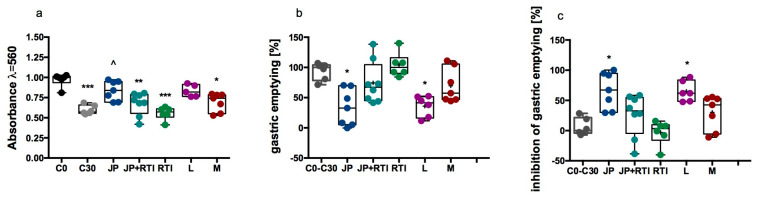
Effects of JP-14, RTI-7470-44, and reference compounds on gastric emptying in mice: absorbance of gastric samples (**a**); % gastric emptying (**b**); % inhibition of gastric emptying (**c**); C0—control sample collected immediately after dye administration (time 0, zero% gastric emptying); C30—control sample collected 30 min after dye administration (time 30, maximum gastric emptying); JP—JP-14, JP+RTI—JP14+RTI, RTI—RTI-7470-44 (antagonist of TAAR1); L—loperamide; M—metformin; mean (marked with “+”) ± SD (box); whiskers indicate minimum and maximum values; median (marked with a line); *n* = 6–8; Kruskal–Wallis’s test; Dunn’s post hoc test; *—vs. C, ^—vs. C30 (**a**); *—vs. C0-C30 i.e., 100% (**b**); *—vs. C0-C30 i.e., 0% (**c**); difference was considered when *,^ *p* < 0.05, ** *p*< 0.01, *** *p* < 0.001.

**Figure 13 ijms-26-10033-f013:**
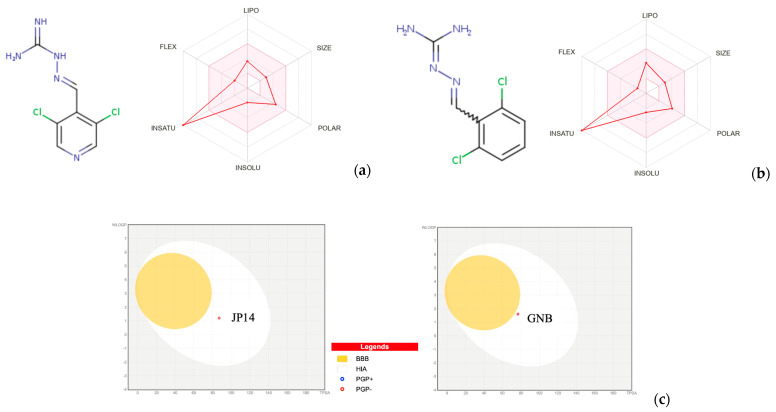
Bioavailability radars for JP-14 (**a**) and guanbabenz (**b**); boiled egg plots (**c**). POLAR—polarity, INSOLU—solubility, INSATU—saturation, FLEX—flexibility, LIPO—lipophilicity; the pink area illustrates the desired properties of the molecule; BBB—blood–brain barrier; HIA—intestinal absorption; PGP+—P-glycoprotein substrate; PGP−—P-glycoprotein non-substrate; GNB—guanabenz; TPSA—topological polar surface area; WLOGP—Log P value according to the Wildman and Crippen model—the partition coefficient between n-octanol and water; the outer gray region indicates compounds with reduced gastrointestinal absorption and restricted BBB penetration, the white region signifies the physicochemical space of molecules that are most likely to undergo passive human intestinal absorption (HIA), and the yellow region denotes the physicochemical space of molecules with the greatest likelihood of crossing the BBB.

**Figure 14 ijms-26-10033-f014:**
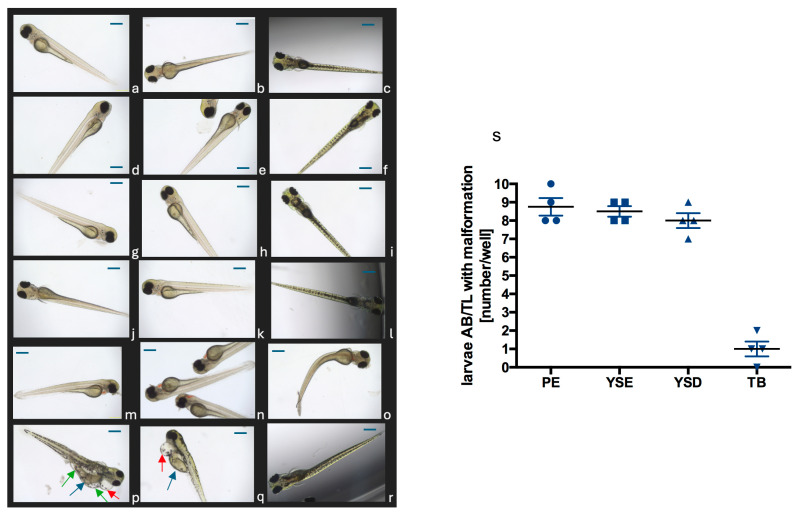
Representative pictures of *Danio rerio* larvae after incubation with JP-14 (toxicity model) and quantitative analysis of disturbances. Control larva (Casper)—incubated with E3 alone (**a**,**b**), control larva (ABTL)—incubated with E3 alone (**c**), larvae (Casper or ABTL) incubated with JP-14 at concentration 5 μg/mL (**d**–**f**), larvae (Casper or ABTL) incubated with JP-14 at concentration 10 μg/mL (**g**–**i**), larvae (Casper or ABTL) incubated with JP-14 at concentration 25 μg/mL (**j**–**l**), larvae (Casper) incubated with JP-14 at concentration 50 μg/mL, all dead (**m**–**o**), larvae (ABTL) incubated with JP-14 at concentration 50 μg/mL (**p**–**r**), quantitative analysis of disorders in group of AL/TL larvae exposed to a concentration of 50 µg/mL of JP-14 (**s**). PE—pericaldial edema (red arrow), YSE—york sac edema (green arrow), YSD—york sac deformation (blue arrow), TB—tail bend; magnification 4×, a scale bar representing 500 µm is shown in each image.

**Figure 15 ijms-26-10033-f015:**
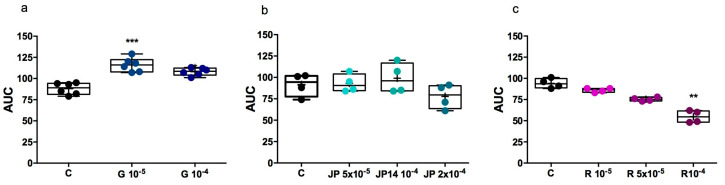
Effects of guanabenz (**a**), JP-14 (**b**), and RTI-7470-44 (**c**) on in vitro whole blood aggregation induced by collagen (1.6 µg/mL). C—control, G—guanabenz, JP—JP-14, R—RTI-7470-44; mean (marked with “+”) ± SD (box); whiskers indicate minimum and maximum values; median (marked with a line); *n* = 4–6; Kruskal–Wallis’s test; Dunn’s post hoc test; ** *p* < 0.01, *** *p* < 0.001 versus control group (0.1% DMSO in saline); AUC—area under the curve.

**Table 1 ijms-26-10033-t001:** Functional activities at TAAR1 and off-target receptors (5-HT_2C_, α_1_, α_2A-B,_ -adrenoceptors).

Compd	TAAR1 Ago E_max_/EC_50_ [µM]	TAAR1 Ant EC_50_ [nM]	5-HT2C Ago EC_50_ [nM]	5-HT2C Ant EC_50_ [nM]	α_1_ *K*_i_ [nM]	α_2_ *K*_i_ [nm.]	α_2A_ Ago E_max_/EC_50_ [nM]	α_2B_ Ant IC_50_ [nM]
10 (JP-14)	11.29 ± 1.4	n.a.	n.a.	n.a.	n.a.	126.0 ± 27.0	15%	622
GNB	60%/ 0.301 ± 0.04	n.a.	n.a.	n.a.	n.t.	2.6 ± 0.6	16.32	n.a.
4-OH GNB	34%/ (10^−4^M) ^a^	n.t.	n.t.	n.t.	n.t.	19.4	316.3 Ago	330.2 Ant
TYR	5.87 ± 0.47	n.t.	n.t.	n.t.	n.t.	n.t.	n.t.	n.t.
RTI	n.a.	167.6 ± 50.7	n.t.	n.t.	n.t.	n.t.	n.t.	n.t.
5-HT	n.a.	n.a.	0.8 ± 0.1	n.t.	n.t.	n.t.	n.a.	n.t.
MSE	n.a.	n.a.	n.t.	1.1 ± 0.1	n.t.	n.t.	n.a.	n.t.
PHT	n.t.	n.t.	n.t.	n.t.	10.9 ± 0.8	n.t.	n.t.	n.t.
CLO	n.t.	n.t.	n.t.	n.t.	n.t.	3.1 ± 0.4	n.t.	n.t.
BRIM	n.t.	n.t.	n.t.	n.t.	n.t.	n.t.	7.25	n.a.
YOH	n.t.	n.t.	n.t.	n.t.	n.t.	n.t.	n.a.	5.43 ± 1.8

means ± SD; n.a.—no activity at 10–5 M; n.t.—not tested; GNB—guanabenz; TYR—tyramine, RTI—RTI-7470-44; 5-HT—serotonin; MSE—methysergide; PHT—phentolamine; CLO—clonidine; BRIM—brimonidine; YOH—yohimbine; ^a^—[7].

**Table 2 ijms-26-10033-t002:** Physicochemical, pharmacokinetic, and drug-likeness properties of JP-14 compared to guanabenz.

Properties		JP-14	Guanabenz
Molecular weight		232.07 g/mol	231.08 g/mol
Number heavy atoms		14	14
Number aromatic heavy atoms		6	6
Fraction Csp3		0.00	0.00
Number H-bond acceptors		3	2
Number H-bond donors		3	2
Number rotatable bonds		3	2
Molar refractivity		56.81	59.02
TPSA		87.15 Å^2^	76.76 Å^2^
Lipophilicity	Log Po/w (iLOGP)	0.65	1.57
	Log Po/w (WLOGP)	1.21	1.60
	Log Po/w (MLOGP)	0.70	2.22
	Log Po/w (XLOGP3)	0.88	1.73
Water Solubility	Log S (ESOL)	−1.95	−2.55
	Solubility	2.59 × 10^0^ mg/mL 1.12 × 10^−2^ mol/L	6.55 × 10^−1^ mg/mL 2.83 × 10^−3^ mol/L
	Class	very soluble	soluble
	Log S (Ali)	−2.29	−2.96
	Solubility	1.18 × 10^0^ mg/mL 5.08 × 10^−3^ mol/L	2.54 × 10^−1^ mg/mL 1.10 × 10^−3^ mol/L
	Class	Soluble	soluble
Pharmacokinetics	GI absorption	High	High
	Intestinal absorption (human)	69.295%	88.791%
	BBB permeant	No	No
	P-gp substrate	No	No
	Log Kp	−7.09 cm/s	−6.48 cm/s
	Bioavailability score	0.55	0.55
	Volume of distribution at steady state	0.420 log L/kg	0.928 log L/kg
	Fraction unbound (human)	0.687 Fu	0.416 Fu
	Inhibitor: CYP1A2, CYP2C19, CYP2C9, CYP2D6, CYP3A4	No No	Yes No
	Total clearance	0.376 log ml/min/kg	0.209 log ml/min/kg
Lipinski filter		Yes, 0 violation	Yes, 0 violation
Synthetic accessibility score		2.55	2.30

TPSA—topological polar surface area [46]; iLOGP [47]; WLOGP [48]; MLOGP [49]; ESOL [50]; Ali [51]; XLOGP3—LogP calculated by XLOGP3 program; GI—gastrointestinal; BBB—blood–brain barrier; P-gp—P-glycoprotein; Log Kp—skin permeation [52]; Bioavailability score—probability that a compound will have F > 10% in the rat [53]; Lipinski filter—druglike properties according to Lipinski’s rule: molecular weight < 500 Da, H-bond donors (OH and NH) ≤ 5, H-bond acceptors (N and O) ≤ 10, and MLOG *p* ≤ 4.15 [49]; Synthetic accessibility score: from 1 (very easy) to 10 (very difficult).

**Table 3 ijms-26-10033-t003:** Toxicity of JP-14 compared to guanabenz.

Toxicity Prediction	JP-14	Guanabenz	Unit
Max. tolerated dose (human)	0.625	0.357	Numeric (log mg/kg/day)
Oral Rat Acute Toxicity (LD_50_)	2.968	2.801	Numeric (mol/kg)
Oral Rat Chronic Toxicity (LOAEL)	1.525	2.498	Numeric (log mg/kg b.w./day)
Hepatotoxicity	No	No	Categorical (Yes/No)
Skin Sensitisation	Yes	Yes	Categorical (Yes/No)
hERG I inhibitor	No	No	Categorical (Yes/No)
hERG II inhibitor	No	No	Categorical (Yes/No)
AMES toxicity	Yes	Yes	Categorical (Yes/No)

hERG—human ether-à-go-go-related gene (a human gene encoding a subunit of a potassium channel important for cardiac repolarization).

## Data Availability

The data are included in the manuscript; detailed information is available from the corresponding authors.

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
