# Peer review of "JP-14: A Trace Amine-Associated Receptor 1 Agonist with Anti-Metabolic Disorder Potential"

_ijms, 2025, doi:10.3390/ijms262010033_

Round 1
Reviewer 1 Report
Comments and Suggestions for Authors
The overall quality of the data presentation in this manuscript needs improvement. Please see the specific comments listed as follows:
- Compound dosing: The co-treatment groups used the full-dose combination of each compound. However, the authors did not provide dose-response data for individual compounds to justify that the selected doses are optimal. In addition, full-dose combinations may increase toxicity; therefore, testing half-dose combinations would strengthen the conclusions.
- Figure 7: The DAPI channel appears very dark. Please increase the brightness of the DAPI signal to improve visualization.
- Figure 10: Quantification for the Control group should be included to allow appropriate comparisons.
- Figure 11: Panel a lacks the control group (E3). Moreover, Figure 16 should be integrated into Figure 11 to provide representative images of the yolk ROI area.
- Figure 12: Panels b and c have x-axis labels that are not consistent with panel a or with the figure legend. This should be corrected for clarity.
- Larvae images: All larvae images, including those in Figures 10, 14, and 16, lack scale bars. Scale bars are necessary for proper interpretation and should be added.
Reviewer 2 Report
Comments and Suggestions for Authors JP-14 as a novel aminoguanidine-based TAAR1 agonist can enhance glucose uptake in HepG2 cells and attenuate lipid accumulation during adipocyte differentiation in 3T3-L1 cells. In differentiated 3T3-L1 adipocytes, JP-14 can reduce intracellular levels of both neutral lipids and phospholipids to show dual anti-steatotic and anti-phospholipidotic activity, promote lipid mobilization and partially prevented fructose-induced lipid accumulation to demonstrate systemic metabolic benefits in vivo, markedly delay gastric emptying in mice, indicate that JP-14 is a safe and metabolically active TAAR1 agonist with multifaceted effects on glucose and lipid metabolism. These results are interesting to audiences. After English polishing, this manuscript can be accepted for publication based on my view. Comments on the Quality of English LanguageThe English of this manuscript needs to be polished.
Reviewer 3 Report
Comments and Suggestions for Authors
Results and discussion:
Comment 1: Please specify in the figure legend 7: e.g. DAPI, FITC, TRITC.
Comment 2: You mention the following in your Results section: “Persistent phospholipidosis can interfere with normal lysosomal function, membrane dynamics, and cell signaling [26]”. Please support this idea with additional evidence and include another example to enhance clarity.
Comment 3: I suggest adding more information about 'JP-14' as the main candidate with the most favorable metabolic profile, as relevant background prior to section 2.2.4. Influence of JP-14 on lipid utilization in zebrafish model.
Comment 4: “JP-14 and guanabenz are not defined as substrates for P-glycoprotein which is advantageous from the point of view of potential use them as drugs, because P-glycoprotein is an efflux transporter that can pump drugs out of cells, which may cause a failed treatment". Please provide stronger support for this statement, as it is relevant and requires greater clarity.
Comment 5: Please check and improve the quality of some figures, as several appear pixelated and contain important information.
Comment 6: Table 3. Please add 'hERG' in the table legend.
Comment 7: Please clarify the limitations and future perspectives of your study, as well as highlight the importance of the results and their potential benefits and applications.
Material and methods:
Comment 8: The experiment was conducted according to Miyasaka et al. and Kotańska et al., with some minor modifications. Could you specify what minor modifications you are referring to?
Comment 9: The experimental procedures were performed from 10:00 a.m. to 14:00 p.m. Could you clarify on which dates the experiments were conducted?
Comment 10: I appreciate the supplementary material; it provides important information. However, I suggest improving its presentation (e.g., by using one table per sheet) and uploading it in PDF format to avoid changes in formatting style.
Reviewer 4 Report
Comments and Suggestions for Authors
This article systematically reports the pleiotropic effects of a novel TAAR1 agonist, JP-14, on metabolic regulation, including enhanced glucose uptake, suppressed lipid accumulation, and delayed gastric emptying. The article also preliminarily validates its TAAR1-mediated mechanism. The study design is comprehensive, encompassing molecular docking, in vitro cell experiments, and zebrafish and mouse models. The data are well-supported, demonstrating considerable innovation and translational potential. However, the article still leaves room for improvement in mechanistic depth, rigorous data interpretation, and some experimental designs, and major revisions are recommended.
1. JP-14 exhibited a maximal effect greater than that of tyramine in the cAMP assay. The authors speculate that non-TAAR1 mechanisms (such as PDE inhibition or direct AC activation) may be involved, but further verification is not possible. Using TAAR1-KO cells or siRNA knockdown of TAAR1 is recommended to verify whether JP-14 still elevates cAMP.
2. JP-14 exhibits 15% partial agonist activity at αâ‚‚A-AR, but its role in metabolism has not been clearly verified. The effects of JP-14 on insulin secretion can be evaluated in cells with high αâ‚‚A-AR expression (e.g., HT-29 cells) or pancreatic β-cells.
3. Although JP-14 is predicted to be unable to cross the blood-brain barrier, its homolog, guanabenz, does enter the brain, which requires experimental verification.
4. The concentration used in the zebrafish experiments (10 μg/mL) has not been converted to or justified by the in vitro ECâ‚…â‚€ value (11 μM).
5. Error bars in some figures and tables are inconsistent (SD vs. SEM), and individual data points are not presented.
6. A comparison with known TAAR1 agonists (e.g., ulotaront) should be added to the Discussion section. The potential applications and limitations of JP-14 in treating metabolic syndrome should be explored.
This article systematically reports the pleiotropic effects of a novel TAAR1 agonist, JP-14, on metabolic regulation, including enhanced glucose uptake, suppressed lipid accumulation, and delayed gastric emptying. The article also preliminarily validates its TAAR1-mediated mechanism. The study design is comprehensive, encompassing molecular docking, in vitro cell experiments, and zebrafish and mouse models. The data are well-supported, demonstrating considerable innovation and translational potential. However, the article still leaves room for improvement in mechanistic depth, rigorous data interpretation, and some experimental designs, and major revisions are recommended.
1. JP-14 exhibited a maximal effect greater than that of tyramine in the cAMP assay. The authors speculate that non-TAAR1 mechanisms (such as PDE inhibition or direct AC activation) may be involved, but further verification is not possible. Using TAAR1-KO cells or siRNA knockdown of TAAR1 is recommended to verify whether JP-14 still elevates cAMP.
2. JP-14 exhibits 15% partial agonist activity at αâ‚‚A-AR, but its role in metabolism has not been clearly verified. The effects of JP-14 on insulin secretion can be evaluated in cells with high αâ‚‚A-AR expression (e.g., HT-29 cells) or pancreatic β-cells.
3. Although JP-14 is predicted to be unable to cross the blood-brain barrier, its homolog, guanabenz, does enter the brain, which requires experimental verification.
4. The concentration used in the zebrafish experiments (10 μg/mL) has not been converted to or justified by the in vitro ECâ‚…â‚€ value (11 μM).
5. Error bars in some figures and tables are inconsistent (SD vs. SEM), and individual data points are not presented.
6. A comparison with known TAAR1 agonists (e.g., ulotaront) should be added to the Discussion section. The potential applications and limitations of JP-14 in treating metabolic syndrome should be explored.
Round 2
Reviewer 1 Report
Comments and Suggestions for Authors
The toxicity study relies on an AI-based tool to predict toxicity. While this is an interesting approach, experimental validation is essential to confirm these predictions. Additionally, in Figure 7, some of the single-channel images do not align correctly with their corresponding merged images. Please correct these inconsistencies.
Author Response
Comment 1:
The toxicity study relies on an AI-based tool to predict toxicity. While this is an interesting approach, experimental validation is essential to confirm these predictions.
Response 1:
We certainly agree that an AI-based tool to predict toxicity is insufficient for toxicological studies, although very helpful in the first phase of research about drug discovery, as it allows us to avoid testing compounds that could be highly toxic and thus avoid spending money on toxic therapeutic approaches.
Therefore, we performed preliminary toxicity studies based on an AI-based tool, as well as in vitro preliminary cytotoxicity assessment of test compound (see 2.4.2., page19 and suplemmentary data) and in vivo toxicity tests of various concentrations of our compound JP-14 in the zebrafish model (see 2.4.3., pages 20-21).
In subsequent studies, we also plan to conduct other studies, such as determining developmental toxicity in a fish model, the effect on heart function, and permeability across the blood-brain barrier. We mention these plans in lines 686-690 on page 22 of our manuscript.
Comment 2:
Additionally, in Figure 7, some of the single-channel images do not align correctly with their corresponding merged images. Please correct these inconsistencies.
Response 2:
We apologize for this oversight; when correcting the DAPI channel in the previous round, we forgot to update the merged channel. The correct image has now been inserted (see Figure 7).
Reviewer 4 Report
Comments and Suggestions for Authors
The authors have responded to the reviewers' concerns and revised the manuscript, which is acceptable.
Comments on the Quality of English LanguageThe authors have responded to the reviewers' concerns and revised the manuscript, which is acceptable.
Author Response
Comments 1:
The authors have responded to the reviewers' concerns and revised the manuscript, which is acceptable.
Responce 1:
On behalf of all the authors, thank you for this comment.
Round 3
Reviewer 1 Report
Comments and Suggestions for Authors
In Figure 7, several single-channel images appear misaligned with their corresponding merged images. For example, the TRITC channel does not match the merged image in the DMSO, Tyramine, Tyramine + RTI-7470-44 groups, and the FITC channel does not align properly with the merged image in the Tyramine, JP-14, RTI-7470-44, Tyramine + RTI-7470-44, JP-14 + RTI-7470-44 groups. Please review and correct these inconsistencies to ensure accurate image representation.
Author Response
Comments 1:
In Figure 7, several single-channel images appear misaligned with their corresponding merged images. For example, the TRITC channel does not match the merged image in the DMSO, Tyramine, Tyramine + RTI-7470-44 groups, and the FITC channel does not align properly with the merged image in the Tyramine, JP-14, RTI-7470-44, Tyramine + RTI-7470-44, JP-14 + RTI-7470-44 groups. Please review and correct these inconsistencies to ensure accurate image representation.
Responce 1:
Thank you for this valuable comment. We have carefully reviewed Figure 7 and confirmed that a misalignment occurred during the image export and copy process. To ensure the highest accuracy and consistency of image representation, we decided to re-acquire all representative images directly from the High Content Screening system. The revised figure now contains correctly aligned single-channel and merged images.
Round 4
Reviewer 1 Report
Comments and Suggestions for Authors
All my concerns have been fully and thoughtfully addressed in the revised version.